# Epitaxy of wafer-scale single-crystal MoS$_2$ monolayer via buffer layer control

Lu Li[1,2], Qinqin Wang[1,2], Fanfan Wu[1,2], Qiaoling Xu[3,4], Jinpeng Tian[1,2], Zhiheng Huang[1,2], Qinghe Wang[5], Xuan Zhao[1,2], Qinghua Zhang[1,2], Qinkai Fan[1,2], Xiuzhen Li[1,2], Yalin Peng[1,2], Yangkun Zhang[1,2], Kunshan Ji[1,2], Aomiao Zhi[1,2], Huacong Sun[1,2], Mingtong Zhu[1,2], Jundong Zhu[1,2], Nianpeng Lu[1,2,3], Ying Lu[1,2], Shuopei Wang[3], Xuedong Bai[1,2,3], Yang Xu[1,2], Wei Yang[1,2], Na Li[3], Dongxia Shi[1,2,3], Lede Xian[3], Kaihui Liu[5], Luojun Du[1,2] ✉ & Guangyu Zhang[1,2,3] ✉

Monolayer molybdenum disulfide (MoS$_2$), an emergent two-dimensional (2D) semiconductor, holds great promise for transcending the fundamental limits of silicon electronics and continue the downscaling of field-effect transistors. To realize its full potential and high-end applications, controlled synthesis of wafer-scale monolayer MoS$_2$ single crystals on general commercial substrates is highly desired yet challenging. Here, we demonstrate the successful epitaxial growth of 2-inch single-crystal MoS$_2$ monolayers on industry-compatible substrates of *c*-plane sapphire by engineering the formation of a specific interfacial reconstructed layer through the S/MoO$_3$ precursor ratio control. The unidirectional alignment and seamless stitching of MoS$_2$ domains across the entire wafer are demonstrated through cross-dimensional characterizations ranging from atomic- to centimeter-scale. The epitaxial monolayer MoS$_2$ single crystal shows good wafer-scale uniformity and state-of-the-art quality, as evidenced from the ~100% phonon circular dichroism, exciton valley polarization of ~70%, room-temperature mobility of ~140 cm$^2$v$^{-1}$s$^{-1}$, and on/off ratio of ~10$^9$. Our work provides a simple strategy to produce wafer-scale single-crystal 2D semiconductors on commercial insulator substrates, paving the way towards the further extension of Moore's law and industrial applications of 2D electronic circuits.

Since the creation of integrated circuits in the 1960s, silicon transistors, following the Moore's law, have been shrinking to boost performance and reduce costs over the past half a century[1]. Today, as conventional silicon transistors enter the sub-10 nm technology node and approach their physical limits, new channel materials are urgently required to further scale transistors and extend Moore's law beyond silicon electronics[2–5]. Two-dimensional (2D) semiconductors with atomic thicknesses and dangling-bond-free flat surface have attracted tremendous interest and possess promising prospects for scaling transistors to the end of roadmap[3,4,6]. The International Roadmap for Devices and Systems (IRDS) has listed 2D semiconductors as the potential channel materials in 2017, and

[1]Beijing National Laboratory for Condensed Matter Physics, Institute of Physics, Chinese Academy of Sciences, 100190 Beijing, China. [2]School of Physical Sciences, University of Chinese Academy of Sciences, 100049 Beijing, China. [3]Songshan Lake Materials Laboratory, Dongguan 523808 Guangdong, China. [4]College of Physics and Electronic Engineering, Center for Computational Sciences, Sichuan Normal University, Chengdu 610068, China. [5]Collaborative Innovation Center of Quantum Matter and School of Physics, Peking University, 100871 Beijing, China. ✉e-mail: luojun.du@iphy.ac.cn; gyzhang@iphy.ac.cn

forecasts that 2D electronic circuits will be commercially available by 2034[5].

Monolayer molybdenum disulfide ($MoS_2$) has been considered as one of the most promising 2D semiconductor candidates for high performance electronic circuits because of its intrinsic high mobility, excellent gate controllability, high on/off current ratio, ultra-low standby current, small dielectric constant, and good stability[7–14]. Indeed, isolated monolayer $MoS_2$ devices have been successfully demonstrated to perform well at ultra-scaled lengths down sub-1 nm, which are inconceivable in the framework of traditional silicon with scaling gate length limit of ~12 nm[5,9–11,15,16]. Monolayer $MoS_2$ transistors have also been identified by Intel as one of three breakthrough technologies to break the scaling limit of silicon. To realize its full potential and high-end industrial applications, it is of utmost importance and a prerequisite to product wafer-scale monolayer $MoS_2$ single crystals on commercial substrates.

Notably, a general theoretical framework is established recently to guide the growth of wafer-scale single-crystal 2D materials[17,18]. In the light of such guideline that the symmetry group of a substrate should be a subgroup of 2D material, $c$-plane sapphire with $C_{3v}$ symmetry offers an industry-compatible substrate for the epitaxial growth of wafer-scale monolayer $MoS_2$ single-crystals with point group $D_{3h}$ ($C_{3v}$ plus a mirror-reflection symmetry $\sigma_h$)[17–19]. Indeed, wafer-scale single-crystal $MoS_2$ as well as other transition metal dichalcogenides (TMDs) have been epitaxially grown on sapphire substrates by surface step engineering (e.g., controlling the surface step orientation and height)[19–24]. It is noteworthy that such a step engineering strategy can also be applied to epitaxially grow various 2D materials on other substrates[25–31]. However, specially-designed substrates such as deliberately engineered off-cut angles or annealing at harsh temperatures are typically required for surface step engineering. Meanwhile, recent studies demonstrate that growth conditions (e.g., the $S/MoO_3$ precursor ratio) can control the interface such as formation of a specific atomically thin interfacial buffer layer and thus also have a strong modulation on the unidirectional domain alignment[26,32–36]. Currently, growth condition controlling is usually coordinated with substrate surface step engineering to achieve the grown of wafer-scale monolayer TMD single-crystals.

Now, a natural question is: can we achieve the epitaxial growth of wafer-scale single-crystal $MoS_2$ monolayers on industry-compatible $c$-plane sapphire substrates by purely growth condition control without the aid of surface step engineering? In this work, we answer this concern in the affirmative and report the synthesis of 2-inch monolayer $MoS_2$ single crystals on general $c$-plane sapphire substrates by precisely engineering the formation of a specific buffer layer within the substrate-epilayer gap through the control of $S/MoO_3$ precursor ratio. The unidirectional alignment and seamless stitching of $MoS_2$ domains are comprehensively demonstrated via multi-scale characterizations ranging from atomic- to macroscopic-scale. The high quality of as-grown monolayer $MoS_2$ single crystals is evidenced by the state-of-the-art electron, phonon, and exciton properties, comparable to or even better than that of exfoliated ones. Our results offer a simple strategy to epitaxially grow wafer-scale single-crystal $MoS_2$ monolayer on commercial insulator substrates and can also been applied to a wide variety of other 2D materials, laying a strong foundation for 2D electronic circuits to fit into industrial settings.

## Results

### Unidirectional domain alignment by buffer layer control

Figure 1a−c presents the optical micrographs of the as-grown $MoS_2$ triangular domains on $c$-plane sapphire substrates with a major miscut angle (~0.2°) towards M-axis under three representative $MoO_3/S$ precursor ratios: 3.9% (Fig. 1a), 4.5% (Fig. 1b) and 5.1% (Fig. 1c). Remarkably, the degree of unidirectional alignment strongly depends on the $MoO_3/$ S precursor ratio, which is defined as:

$$\rho = \frac{n_{max} - n_{min}}{n_{max} + n_{min}} \quad (1)$$

where the $n_{max}$ ($n_{min}$) corresponds to the number of oriented $MoS_2$ domains in the majority (minority). For 4.5% $MoO_3/S$ precursor ratio, the degree of unidirectional alignment is more than 99%. Note that this value is extracted based on orientation statistics across the entire 2-inch wafer (please see Supplementary Note 1 for microscopy images over a ~1 $mm^2$ area). By contrast, the degree of unidirectional alignment is essentially zero for 3.9% and 5.1% $MoO_3/S$ precursor ratios. Figure 1g shows the degree of unidirectional alignment against the $MoO_3/S$ precursor ratios. Clearly, the degree of unidirectional alignment can be continuously tuned from ~0 to ~ 100% by controlling the $MoO_3/S$ precursor ratio (please see Supplementary Note 2 for more $MoO_3/S$ precursor ratios). This indicates that pure $MoO_3/S$ precursor ratio control can endow with the unidirectional domain alignment. In addition to $c$-plane sapphire substrates with a major miscut angle (~0.2°) towards M-axis, the unidirectional domain alignment has also been achieved by precisely controlling the $S/MoO_3$ precursor ratio on pure $c$-plane sapphires, and $c$-plane sapphire substrates with different major miscut angles towards other axes (please see Supplementary Note 3), indicating the universality of our method. It is noteworthy that for all the $MoO_3/S$ ratios we have investigated (ranging from ~3.83% to ~5.55%), it always belongs to a S-rich condition. Consequently, the shape of $MoS_2$ crystals remains unchanged (please see Supplementary Note 4), in contrast prior work where the drastic changes in S/Mo ratio lead to the shape evolution[37,38].

Upon extending the growth time, the unidirectional oriented $MoS_2$ domains start to merge and eventually coalesce into a continuous monolayer film at ~45 min (Fig. 1d–f). Figure 1j shows the photograph of the as-grown 2-inch monolayer $MoS_2$ wafer on $c$-plane sapphire substrates with a major miscut angle towards M axis. Atomic force microscope (AFM) images taken from different locations across a 2-inch wafer manifest a uniform and wrinkle-free monolayer $MoS_2$ film with a thickness of ~0.7 nm (please see Supplementary Note 5). It is worth stressing that compared to surface engineering techniques employed in previous work that require a special design for the substrate[19–21,25,26], our precursor ratio control strategy holds unique advantages and can fit the $c$-plane sapphire substrates with a major miscut angle (~0.2°) towards M axis, which are mainly supplied on the market and industry-compatible (please see Supplementary Note 6). Although surface steps can also be present in c-plane sapphire substrates with a major miscut angle towards M axis, they are perpendicular to the zigzag edge of triangular $MoS_2$ domains and therefore have no effect on the unidirectional domain alignment (please see Supplementary Note 7 for details).

To understand the underlying growth mechanism, we perform the cross-sectional high-angle annular dark-field scanning transmission electron microscopy (HAADF-STEM) of the as-grown $MoS_2$ with unidirectional domain alignment (Fig. 1i). Importantly, an atomically thin buffer layer is observed between the grown $MoS_2$ layer and the $c$-plane sapphire substrate. By contrast, no buffer layer is formed below the as-grown $MoS_2$ is observed when the degree of unidirectional alignment is ~0 (Supplementary Fig. 11). This suggests that the buffer layer within the substrate-epilayer gap is the key to facilitate the unidirectional epitaxy of the monolayer $MoS_2$ and has been recently demonstrated for unidirectional $MoS_2$ epitaxy on $\beta$-$Ga_2O_3$ (Fig. 1h)[35]. It is noteworthy that for previously reported unidirectional TMDs on sapphire substrates, a buffer layer is also typically observed, but its role is usually ignored[19,20,34].

To further confirm the effect of the buffer layer and its role on unidirectional domain alignment, we remove the as-grown unidirectional $MoS_2$ from $c$-plane sapphire substrate by water-assisted

technique[39,40]. It is noteworthy that such removal process utilizes the water intercalation and would not destroy the buffer layer. Then the sapphire substrates with buffer layers are used to re-grow MoS₂. Note that we also put fresh sapphire substrates in the growth chamber at the same time for control samples. For fresh sapphire substrates, two antiparallel domains appear simultaneously (please see Supplementary Fig. 12). By contrast, unidirectional MoS₂ domains are reproduced on the sapphire substrates with buffer layers (Supplementary Fig. 12). This result strongly demonstrates the key role of buffer layer on the unidirectional domain alignment. By performing X-ray photoelectron spectroscopy and density functional theory calculation, we infer that one possible configuration of the buffer layer is O−Mo−O−Al, with Mo exhibiting a (+5) oxidation state (Supplementary Note 9).

### Seamless stitching of MoS₂ domains

To realize the goal of wafer-scale single-crystal MoS₂ monolayers, seamless stitching of aligned grains must also be satisfied simultaneously, in addition to the unidirectional domain alignment[18,41,42]. To verify the seamless stitching and the absence of grain boundaries, partially merged MoS₂ domains were characterized by atomic-resolution aberration-corrected HAADF-STEM. Figure 2a shows a low-magnification STEM image from the merged area of the two aligned MoS₂ domains. The angle between two merged MoS₂ grains is ~60°. Figure 2b displays six representative atomic-resolution STEM images taken at the corresponding locations marked in

Fig. 2a. The identical lattice orientation without grain boundary continuously across the merging zone strongly evidences the seamless stitching of the MoS₂ domains and therefore single-crystal nature. Additionally, the atomic-resolution STEM images show a honeycomb lattice with $d$-spacing of 0.158 nm and 0.274 nm, corresponding to the (11$\bar{2}$0) and (10$\bar{1}$0) planes of monolayer MoS₂, consistent with previous work[43]. More STEM images of merged areas that support the seamless stitching of MoS₂ domains can be found in Supplementary Note 10.

Polarized second-harmonic generation (SHG), which is highly sensitive to grain boundaries[44,45], is further performed to confirm the seamless stitching of MoS₂ domains at the large scale. Figure 2c shows the representative polarized SHG mapping of two unidirectionally merged domains. Uniform signal is observed with no obvious intensity fluctuations across the merging area. This is strongly distinct to the antiparallel domains, where a dark line appears at the grain boundary (Fig. 2e). Figure 2d presents the polarized SHG mapping of a continuous MoS₂ film consisting of unidirectional aligned domains, showing a uniform distribution of intensity over the entire area. This provides a strong proof of the absence of grain boundaries and seamless stitching, in contrast to result of polycrystalline MoS₂ film (Fig. 2f). Besides, etching experiments with hot water vapors are also performed to verify the seamless stitching of unidirectional aligned MoS₂ domains (Supplementary Note 11).

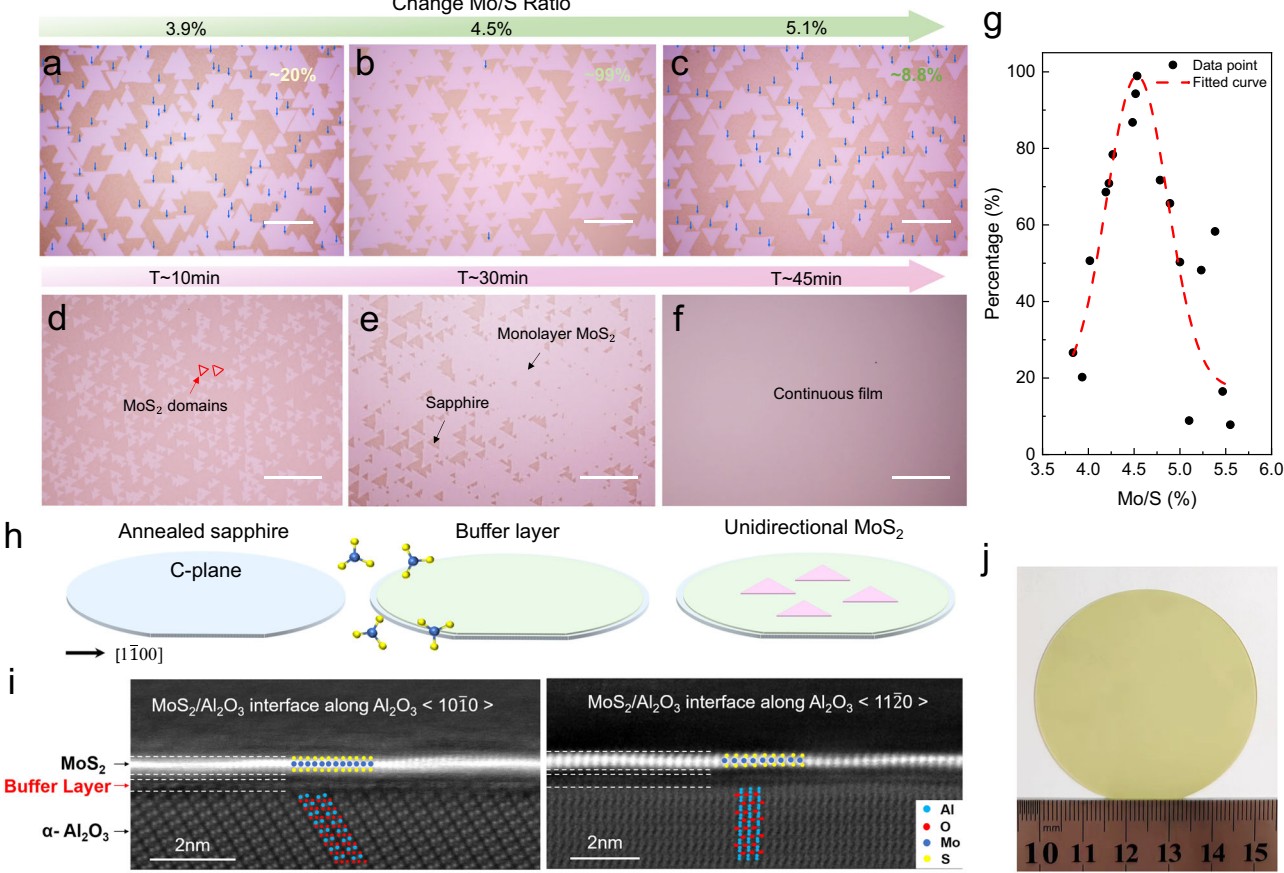

**Fig. 1 | Unidirectional domain alignment enabled by buffer layer control.** Optical microscopy images of the as-grown MoS₂ triangular domains under three representative MoO₃/S precursor ratios: 3.9% (**a**), 4.5% (**b**) and 5.1% (**c**). Scale bar, 20 μm. Optical micrographs of MoS₂ at different growth stages: nucleation (**d**), stitching (**e**) and coalescence of grains (**f**). Scale bar, 30 μm. **g** The degree of unidirectional alignment as a function of the MoO₃/S precursor ratio. Dashed line is the curve fitted by the Gaussian function. **h** Schematic illustration of the buffer layer control strategy toward the synthesis of wafer-scale monolayer MoS₂ single crystals. **i** Cross-sectional high-angle annular dark-field scanning transmission electron microscopy (HAADF-STEM) images of a MoS₂ grown on the *c*-plane sapphire substrate along the <10$\bar{1}$0> and <11$\bar{2}$0> directions. **j** Photograph of the as-grown full-coverage monolayer MoS₂ on 2-inch *c*-plane sapphire substrate with a major miscut angle towards M axis.

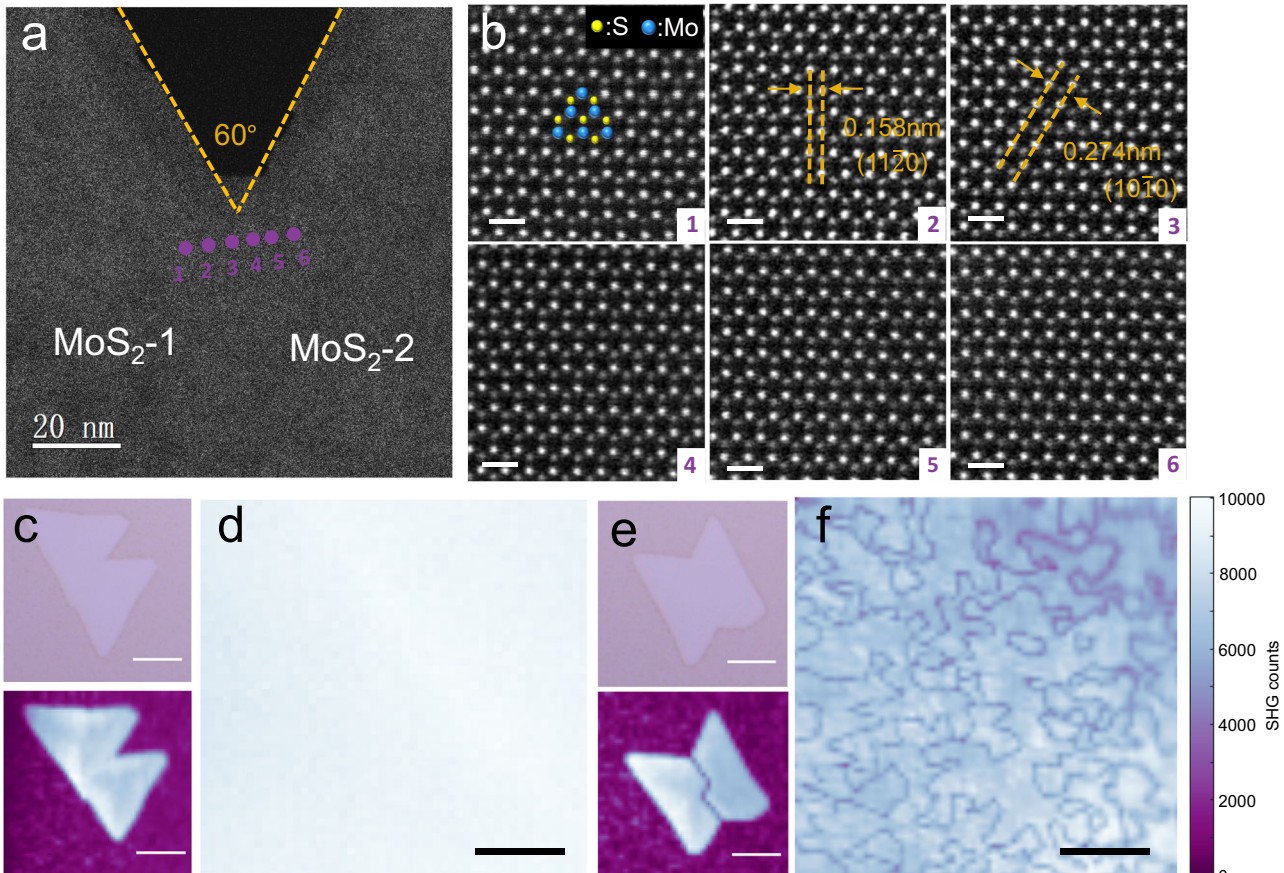

**Fig. 2 | Seamless stitching of unidirectional MoS$_2$ domains. a** HAADF-STEM image of the merging area between two unidirectional MoS$_2$ domains. The orange dashed lines outline the edge of the two aligned MoS$_2$ domains. **b** 1–6: six typically atomic-resolution HAADF-STEM images obtained from the locations marked in **a**, showing that no boundary was formed, the *d*-spacings of the ($10\bar{1}0$) and ($11\bar{2}0$) planes of MoS$_2$ are 0.274 and 0.158 nm, respectively. Scale bars, 0.5 nm. The atomic-resolution HAADF-STEM images are filtered to enhance the contrast. **c, e** Optical microscopy (upper panel) and polarized second-harmonic generation (SHG) mapping (lower panel) of two aligned (**c**) and misaligned (**e**) MoS$_2$ domains. Scale bars, 5 μm. **d, f** Polarized SHG mapping of continuous film formation of single oriented domains (**d**) and misaligned domains (**f**). Scale bars, 20 μm.

## Wafer-scale uniformity and high quality

The quality and uniformity of the as-grown single crystal monolayer MoS$_2$ wafers are illustrated via multi-scale characterizations. Figure 3a presents the low-energy electron diffraction (LEED) patterns measured at twelve random locations across over an area of more than 1 cm$^2$. The essentially identical LEED patterns provide strong evidences of the uniformity of the as-grown single crystal monolayer MoS$_2$. In addition, the LEED patterns show three bright diffraction spots, which unambiguously proves the three-fold rotational symmetry and therefore the single-crystalline nature of the as-grown monolayer MoS$_2$ film (Supplementary Note 12)[21]. Figure 3b is the stacked nine polarization-resolved SHG patterns across a 1 cm$^2$ sample area (SHG patterns from different locations are indicated by different colors). The nearly overlapped SHG six-petal patterns confirm the coherent lattice orientation and uniformity of the as-grown monolayer MoS$_2$ film. Figure 3c, d shows 25 representative room-temperature Raman and photo-luminescence (PL) spectra across a 2-inch MoS$_2$ wafer, respectively (please also refer to Raman and PL mapping in Supplementary Note 13). No apparent changes in the peak position and linewidth of both phonons and excitons are observed, illustrating the wafer-scale uniformity. Figure 3e presents a representative fluorescence microscope image of the as-grown monolayer MoS$_2$. The uniform color contrast confirms the uniformity of the monolayer MoS$_2$ film. The wafer-scale uniformity of the as-grown monolayer MoS$_2$ single crystal is also scrutinized by optical micrographs taken at different locations across a 2-inch range (Supplementary Note 14).

The high quality of the grown 2-inch monolayer MoS$_2$ single crystals is first characterized by aberration-corrected STEM. A statistical analysis of the atomic-resolution images shows that the density of sulfur vacancies in epitaxial monolayer MoS$_2$ is $\sim 5.2 \times 10^{12} cm^2$ (Supplementary Fig. 16), which is an order of magnitude lower than that of the previously reported exfoliated flakes[46]. The quality of the as-grown single crystal monolayer MoS$_2$ film was further scrutinized by helicity-resolved inelastic Raman scattering and exciton valley polarization. Figure 3f shows the circular polarization-resolved Raman spectra at room temperature, excited by left-hand $\sigma^+$ radiation at 532 nm (2.33 eV). Strikingly, the characteristic in-plane phonon $E^1_{2g}$ (out-of-plane vibration $A_{1g}$) features a strong signal only under cross-circularly (co-circularly) polarized detection, while cannot be detected under the co-circularly (cross-circularly) polarized configuration. This evidences the perfect negative (positive) 100% phonon circular dichroism of $E^1_{2g}$ ($A_{1g}$), as expected for the Raman selection rule determined by the perfect structure of monolayer MoS$_2$[47], which proves the negligible defect-lattice scattering and high crystal quality. Figure 3g is the non-polarized PL spectrum of the as-grown monolayer MoS$_2$ at 10 K. The full width at half maximum (FWHM) is ~23 meV, which is only about half of the values in exfoliated flakes[48] (Supplementary Note 15). Figure 3h presents the helicity-resolved PL spectra of the epitaxially grown mono-layer MoS$_2$ at 10 K, excited by $\sigma^+$ radiation on resonance with the A exciton at 633 nm (1.96 eV). Remarkably, the degree of exciton

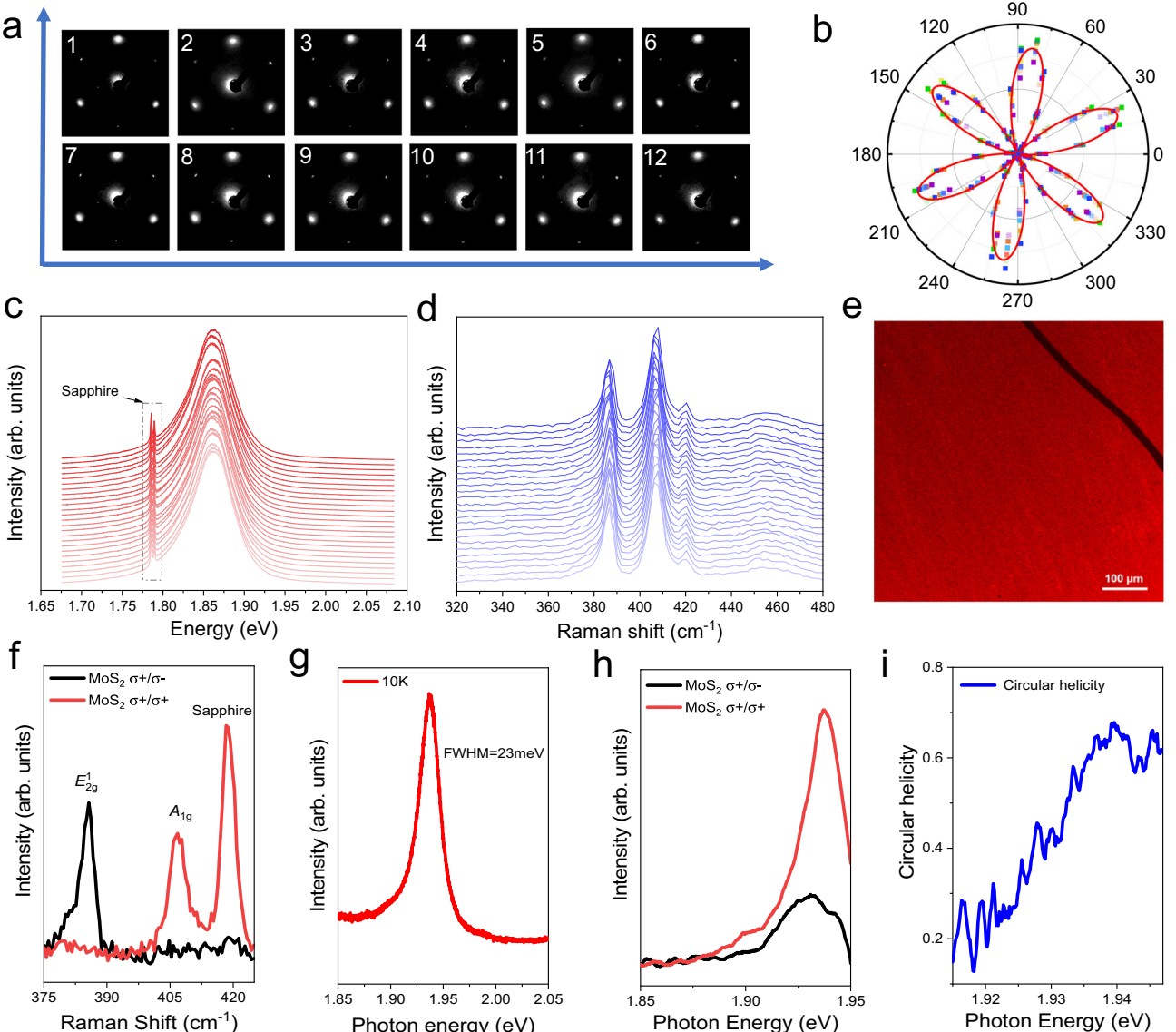

**Fig. 3 | Wafer-scale uniformity and high quality. a** Representative low-energy electron diffraction (LEED) patterns of a continuous MoS$_2$ film indicating the single domain orientation nature of the MoS$_2$ film, taken at 95 eV. **b** Stacked linearly polarized SHG six-petal patterns. SHG from different locations is indicated by symbols with different colors. Solid line is the fitting result to show the six-petal pattern. **c, d** Representative photoluminescence (PL) spectra (**c**) and Raman spectra (**d**) at 25 different locations on the wafer. Dash-dotted box highlights the PL signals from sapphire substrate. **e** Fluorescence microscope image of the as-grown film.

A scratch has been intentionally created in the upper right corner to endow with a clear contrast between MoS$_2$ film and bare sapphire substrate. Scale bars, 100 μm. **f** Polarization-resolved Raman spectra for 532 nm excitation. **g** The low-temperature (10 K) PL spectra of the as-grown MoS$_2$. **h** Circularly polarized PL spectra of the as-grown MoS$_2$ at 10 K, Excitation light is right-handed circularly polarized ($\sigma^+$) at 1.96 eV (633 nm). **i** Circular polarization calculated from the PL spectra in **h**. The high value about 68% indicates the high quality of the MoS$_2$ film.

valley polarization can reach ~68% (Fig. 3i), which is quantified as:

$$\rho_v = \frac{I(\sigma^+/\sigma^+) - I(\sigma^+/\sigma^-)}{I(\sigma^+/\sigma^+) + I(\sigma^+/\sigma^-)} \qquad (2)$$

where $I(\sigma^+/\sigma^+)$ and $I(\sigma^+/\sigma^-)$ denote the exciton emission intensities under the co- and cross-circularly polarized configurations, respectively. Note that the exciton valley polarization of ~ 68% is competitive to or even better than those of the best exfoliated flakes (Supplementary Note 15)[48–50]. Given that exciton valley polarization is highly sensitive to defects as they can introduce intervalley scattering to degrade the circular helicity[48,49], the high exciton valley polarization offers strong evidence for the state-of-the-art quality of our epitaxial monolayer MoS$_2$ single crystals. The high-quality of the as-grown monolayer MoS$_2$ single crystals can be understood from several

aspects. First, we keep a sulfur-rich condition during the growth, facilitating achieving a low density of sulfur vacancies. Second, our chemical vapor deposition (CVD) setup has a unique multisource design (Supplementary Fig. 27), which facilitates the homogeneous cross-sectional source supply and thus high quality. Besides, our monolayer MoS$_2$ single crystals are stitched from unidirectional domains on mono-step sapphire surfaces and the nucleation and growth of domains do not rely on the surface steps, thus we believe that the high-quality of as-grown films would also benefit from such good stitching.

## The state-of-the-art device performances

The high quality of the as-grown single-crystal monolayer MoS$_2$, in principle, would enable the superior device performances. To confirm this, we fabricated field-effect transistors (FETs) for performance

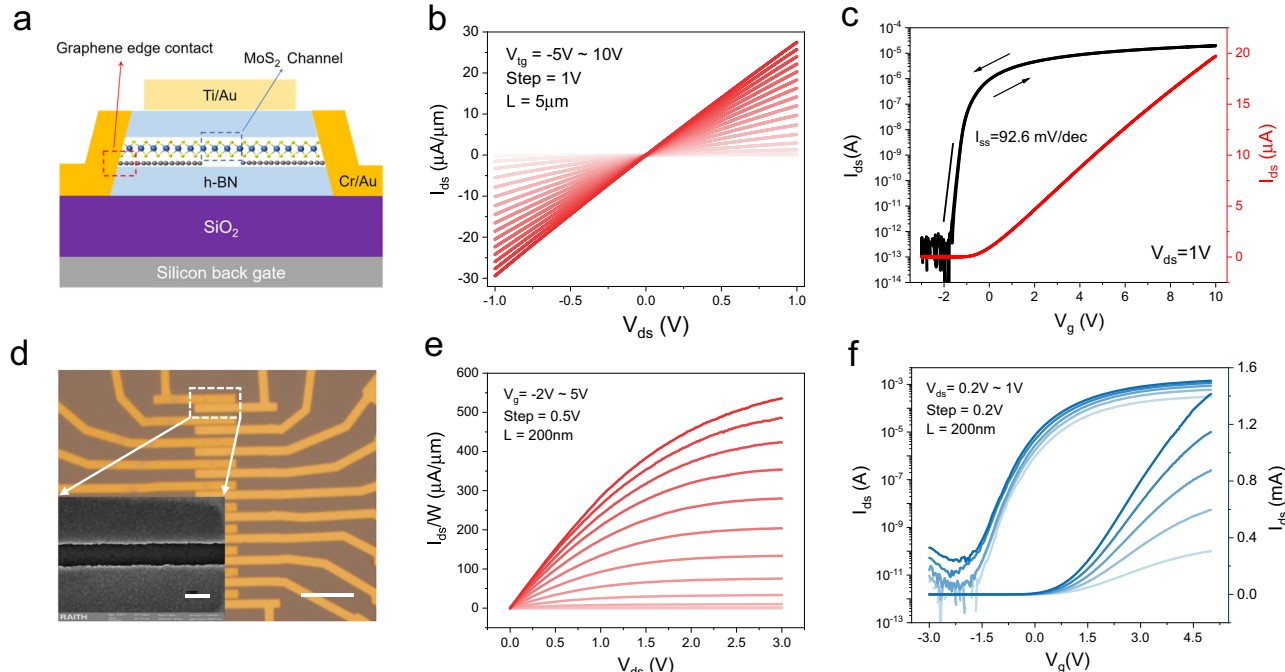

**Fig. 4 | The state-of-the-art device performances. a** Schematic diagram of the typical structure of *h*-BN encapsulated devices. **b, c** Output and transfer curves of a *h*-BN encapsulated MoS₂ device with channel length/width (L/W) of 5 μm/1 μm at V_bg = 0 V. **d** Photograph of the short channel device. Scale bar, 20 μm. Inset shows the scanning electron microscopy image of the device. Scale bar, 200 nm. **e, f** Output and transfer curves of the short channel device with channel length/width (L/W) of 200 nm/5 μm.

benchmark testing. To effectively suppress interfacial scattering and extract intrinsic electronic properties, hexagonal boron nitride (*h*-BN) encapsulated devices based on the epitaxially grown single-crystal monolayer MoS₂ are fabricated utilizing the pick-up technique[51] (see Methods for more details). Figure 4a schematically shows the typical structure of *h*-BN encapsulated devices, with heavily doped silicon (Ti/Au) as the bottom (top) gate. It is noteworthy that monolayer graphene with a narrow electron density distribution around the Fermi level, which can act as a Dirac source and enables the low Schottky barriers[11,52], is adopted as the as source and drain contact electrodes. Figure 4b, c shows the room-temperature output/transfer characteristics of a *h*-BN encapsulated device with channel length/width of 5 μm/1 μm. The thickness of top and bottom *h*-BN layers is ~27 nm and ~34 nm, respectively (Supplementary Fig. 22). Please refer to Supplementary Fig. 24 for the transfer characteristics of more devices. Linear $I_{ds}$-$V_{ds}$ output characteristic curves (Fig. 4b) indicate the ohmic contact behavior. As can be seen from the transfer characteristic curve (Fig. 4c), the device features an on/off ratio up to ~10⁸, negligible electrical hysteresis, and a sharp subthreshold swing of ~92.6 mV/dec across over three orders of magnitude. Remarkably, the extracted field-effect mobility ($\mu = G_m \frac{L}{W C_i V_{ds}}$, where $G_m$ is the transconductance, $C_i$ denotes the gate capacitance, $V_{ds}$ is the source-drain bias, $L$ and $W$ are channel length and width, respectively) can reach ~140 cm²s⁻¹V⁻¹ at room-temperature, which is larger than the results based on poly-crystalline films and competitive to highest value of the exfoliated flakes[39,53–55]. For total 10 *h*-BN encapsulated devices from different locations, the average mobility is ~120 cm²s⁻¹V⁻¹. It is noteworthy that the current electron mobility is underestimated and can be further improved by ultra-low/free contact resistance engineering[54,56–58]. Additionally, batch production of uncapsulated FET arrays over several centimeters is achieved based on the as-grown wafer-scale single-crystal MoS₂ monolayer (Supplementary Note 17). Although the uncapsulated FETs show lower mobility than encapsulated ones (which may stem from interfacial scattering and contaminations in the device fabrication processes), a statistical analysis of 150 FETs over an

inch scale shows a high yield >97% and an average on/off ratio of devices of ~10⁸ (Supplementary Note 17). Finally, 200 nm short-channel FET devices based on the as-grown monolayer MoS₂ were fabricated, as shown in Fig. 4d. Figure 4e, f shows the electrical output and transfer characteristic curves of the short-channel device, respectively. The short-channel device exhibits a saturation on-current density of ~535 μA/μm, a high on/off ratio close to 10⁹ and a sharp subthreshold swing of 107 mV/dec. The overall device performances of our wafer-scale single-crystal MoS₂ monolayers (e.g., mobility of 140 cm²s⁻¹V⁻¹, on-current density of ~535 μA/μm and 10⁹ on/off ratio) are quite competitive to those best results among all the monolayer TMDC transistors reported so far (Supplementary Table 1), holding promising prospects for large-scale integrated applications.

## Discussion

In conclusion, we demonstrate a simple strategy of precursor ratio control for the batch production of wafer-scale single-crystal MoS₂ monolayer films on industry-compatible substrates of *c*-plane sapphire. The epitaxial monolayer MoS₂ single crystals exhibit wafer-scale uniformity and the state-of-the-art properties, evidenced by the perfect phonon circular dichroism, exciton valley polarization of ~70%, room-temperature mobility of ~140 cm²v⁻¹s⁻¹, on-current density of ~535 μA/μm, and a nearly 10⁹ on/off ratio. Our work offers a novel insight into the synthesis of wafer-scale high-quality 2D semiconductors and lays a solid foundation for industrial applications of 2D integrated electronic circuits.

## Methods

### Growth of single-crystal MoS₂ monolayers on *c*-plane sapphire
The growth of single crystalline monolayer MoS₂ was carried out in a three-temperature zone CVD system with multisource design manufactured by Dongguan Join Technology Co., Ltd (please refer to Supplementary Figs. 27 and 28 for the schematic diagram and photographs of our CVD setup, repectively). For a typical growth, the center mini-tube is loaded with sulfur (Alfa Aesar, 99.9%, 6 g) and

flowed with 40 sccm Ar. Note that to effectively load sulfur, a large rectangular chamber is designed ~5 cm from the end of the quartz tube. The surface area of melted sulfur determined by the rectangular chamber is around 25 cm$^2$. For the outside six mini-tubes, every-other-tube is loaded with $MoO_3$ (Alfa Aesar, 99.999%) and flowed with $Ar/O_2$ (40/0.5 sccm); and the other three empty tubes are also flowed with $Ar/O_2$ (40/0.5 sccm). In addition, to avoid the powder being blown away by the carrier gas, we use $MoO_3$ thin flack (the thickness is ~0.5 mm) which is obtained by pressing the $MoO_3$ powder (Alfa Aesar, 99.999%) with a hydraulic press. All the quartz tubes have an inner diameter of ~1 cm. The distance between $MoO_3$ and substrate (sulfur) is ~32 cm–40 cm (~17 cm). During the growth, sulfur, $MoO_3$ and sapphire substrate are placed at first, second, and third temperature zones, respectively. The temperatures for the sulfur, $MoO_3$, and sapphire substrate are 120, 560, and 880 °C, respectively. The pressure is kept at about 1 torr during the growth process. The growth time ranged from 15 mins (to obtain single oriented discrete $MoS_2$ domains) to 45 mins (to obtain single oriented continuous $MoS_2$ films), and the pressure was kept at about 1 torr during the growth process, and the LPCVD system was cooled to room temperature in the argon gas stream after the growth.

AFM imaging was performed by Asylum Research Cypher S. The thickness of the grown $MoS_2$ film is measured by scraping the film at the wafer corner. LEED measurements were performed in vacuum environments ($<3 \times 10^{-7}$ Pa) using the Omicron LEED system. A high-resolution four-circle X-ray diffractometer (Smartlab, Rigaku) was used to characterize the in-plane crystalline symmetry of the grown wafer-scale $MoS_2$ film.

## Optical characterizations

SHG microscopy measurements was performed in a homemade system. A femtosecond laser with 780 nm central wavelength (~100 fs, 100 MHz, generated by MenloSystems) and power of 1.8 mW was used to excite the sample. The laser beam passes through a linear polarizer and was tightly focused to ~1 μm spot diameter by a 100× objective (NA = 0.9, Nikon). The $MoS_2$ samples were attached to a piezoelectric stage to realize sub-micrometer scanning. The SHG signals were collected by the same objective, then passed through another linear polarizer whose polarization angle is perpendicular to the excitation beam. A band pass filter was used to filter out the 780 nm fundamental beam. The SHG intensities were measured by a spectrometer (Princeton Instrument) with an exposure time of 0.3 s each pixel of the mapping.

Both Raman and PL spectra were collected using a HORIBA spectrometer (LabRAM HR Evolution) in a confocal backscattering configuration. For room-temperature Raman spectra, off-resonance light from 532 nm (2.33 eV) continuous laser with a power of about 900 μw was focused through a Nikon objective (N.A. = 0.5 W.D. = 10.6 F.N. = 26.5) onto the sample with a spot diameter of ~1.5 μm. For PL spectra, on-resonance light from 633 nm (2.33 eV) continuous laser with a power of about 400 μw was focused through the same Nikon objective onto the sample with a spot diameter of ~1 μm. The sample was placed in an optical chamber with a high vacuum and then cooled down to 10 K by a closed cryocooler (CS-204PF-DMX-20B-OM from ARS) for cryogenic PL measurements.

## TEM characterizations

The $MoS_2$ samples for TEM characterizations were prepared by transferring $MoS_2$ onto the TEM grids (Zhongjingkeyi) using the polymethyl-methacrylate-based transfer method. STEM was performed by an aberration-corrected JEM ARM200F (JEOL) at 200 kV and an aberration-corrected JEOL Grand ARM 300 CFEG operated 80 kV. The SAED was performed with a TEM (Philips CM200) operating at 200 kV.

## Transfer of $MoS_2$ films

Propylene carbonate (PC, Chloroform solution of 6% propylene carbonate) was used to pick up boron nitride (h-BN) that has been mechanically exfoliated onto silicon oxide. The picked h-BN was dropped onto the as-grown $MoS_2$ to pick up the single crystal $MoS_2$. Then, two parallel graphene samples were picked up as contact electrodes. Finally, the sample falls onto bulk h-BN on 300-nm silicon oxide substrate to obtain van der Waals heterojunction with clean interface and free of organic residues.

## FET fabrication and electrical measurements

Fabrication of h-BN encapsulated devices. h-BN/$MoS_2$/Graphene/h-BN/$SiO_2$ samples were fabricated via a propylene-carbonate-based transfer method. Ti(2 nm)/Au(30 nm) metal electrodes were deposited as the top gate of the device using electron beam lithography (EBL), electron beam evaporation and lift-off techniques. EBL was used to reverse expose the Hall shape of the device, and reactive ion etching (RIE, Plasma Lab 80 Plus, Oxford Instruments) was used to etch off the excess h-BN/$MoS_2$/Graphene/h-BN to avoid the conduction between the device electrodes. Finally, the device was completed by depositing Cr (3 nm)/Au(30 nm) using an electron beam evaporation system to form a one-dimensional contact with graphene.

Batch fabrication of devices. The patterned Ti (2 nm)/Au (10 nm) metal layer were deposited on sapphire substrate as gate electrodes by UV lithography (MA6, Karl Suss) and electron beam evaporation system, followed by deposition of 20 nm $HfO_2$ as the dielectric layer by atomic layer deposition (ALD). The as-grown single-crystal $MoS_2$ was transferred onto $HfO_2$/Au/Ti/sapphire substrate and then patterned into channels by UV lithography and RIE. The source-drain electrodes are Au/Ti/Au (2/2/10 nm), which are defined by UV lithography, electron beam evaporation and lift-off processes.

Fabrication of short channel devices. The as-grown single-crystal $MoS_2$ was transferred to a $HfO_2$ (5 nm)/Si substrate, and the dielectric layer was deposited by ALD. EBL and RIE were then used to defined the channel and drain-source, and Au/Ti/Au (2/2/10 nm) were deposited as drain-source electrodes.

Electrical measurements were carried out with an Agilent B1500 semiconductor parameter analyzer in a four-probe vacuum station with a bass pressure of ~$10^{-6}$ mbar at room temperature.

## Data availability

Relevant data supporting the key findings of this study are available within the article and the Supplementary Information file. All raw data generated during the current study are available from the corresponding authors upon request.

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

## Acknowledgements

We thank Yi Wan for valuable discussions. This work is supported by the National Key Research and Development Program of China (Nos. 2021YFA1202900, 2021YFA1400502, 2021YFA1401300, 2023YFA1407000, 2022YFA1402503), the Key-Area Research and Development Program of Guangdong Province, China (Grant No. 2020B0101340001), Guangdong Major Project of Basic and Applied Basic Research (2021B0301030002), the National Science Foundation

of China (NSFC) under Grant Nos. 61888102, 12274447, 12074412 and 62204166, the Strategic Priority Research Program of Chinese Academy of Sciences (CAS) under Grant No. XDB0470101.

## Author contributions

G.Z. and L.D. designed the research; L.L. performed the growth, high-resolution AFM imaging, TEM, spectroscopic characterizations, device fabrication, and electrical transport measurements with help from Q.W. and Y.Z.; F.W. and J.Z. fabricated the h-BN encapsulated devices; J.T. and X.L. fabricated the short-channel devices; Y.P. fabricated batch devices; Z.H. and K.J. performed the cryogenic PL measurements; Q.H.W. and K.L. performed LEED; Q.Z., A.Z., H.S. and X.B. performed TEM; X.Z. and Y.X. performed SHG; M.Z. and N.P.L. performed XRD; Q.F. and Y.L. performed fluorescence microscope imaging; Q.X. and L.X. performed the theoretical calculations; L.L., Y.Z., W.Y., D.S., S.W., N.L. L.D., and G.Z. analyzed data; L.L., L.D., and G.Z wrote and all authors commented on the manuscript.

## Competing interests

The authors declare no competing interests.
