## [Peer Review File · Nature Communications]

Epitaxy of wafer-scale single-crystal MoS₂ monolayer via
buffer layer controlEditorial Note: Parts of this Peer Review File have been redacted as indicated to remove third-party material where no permission to publish could be obtained.

REVIEWER COMMENTS

Reviewer #1 (Remarks to the Author):

Comments:

The manuscript “Epitaxy of wafer-scale single-crystal MoS₂ monolayer via kinetic control” reported the unidirectional growth of MoS₂ and 2-inch single crystals on the general c-plane sapphire substrate via controlling the growth kinetics. This is an important research topic and attracts wide attention. The reviewer finds the results interesting but lacks a systematic study of the growth mechanism to support the innovation. Regarding the unidirectional MoS₂ growth, some important work in this field has raised their representative viewpoints, such as surface steps (<https://doi.org/10.1038/s41586-022-04523-5>), surface symmetry (<https://doi.org/10.1038/s41565-023-01445-9>), and surface chemistry (<https://doi.org/10.1038/s41565-023-01456-6>). The authors proposed that the kinetic control, i.e. a MoO₃/S ratio, plays a key role, which is possibly reasonable but lacks strict proof and deep-in discussion on the mechanisms. Following are some comments that the reviewer suggests the authors consider.

1. The authors emphasize the kinetics control in the MoS₂ growth, but the reviewer does not find any detailed information/discussion about the kinetics, except simply mentioning the MoO₃/S ratio. What is the kinetics when we talk about it? How does the MoO₃/S ratio affect the growth kinetics? Why does the MoO₃/S ratio affect the domain alignment? Do the different MoO₃/S ratios produce different species contributing to the nucleation and growth, or produce different surface termination that affects the alignment?

2. As claimed in a very recently published work in Nature Nanotechnology (<https://doi.org/10.1038/s41565-023-01445-9>), the bi-steps would produce unidirectional alignment due to surface symmetry, independently from the step direction. The authors should make a clear demonstration different from that work. For this purpose, the substrate characterizations about the step height, and orientation should be supplemented: 1) XRD to check the miscut direction and miscut angle of the substrate (with unidirectional MoS₂ on it), 2) AFM characterization of surface step height (H) and terrace width (W), which should accordance with the XRD data ($\tan\theta=H/W$). Only mono-, non-M-steps are proved, the claimed unidirectional growth on the “general” substrate could be convincing.

3. A recent work published in Nature Communications (<https://doi.org/10.1038/s41467-023-36286-6>) from the same united groups claimed universal epitaxy of single crystal TMDs on sapphire by controlling the simultaneous formation of grain nuclei and substrate steps, where step direction is not a requisite. Does the present work follow the same mechanism as that one? If not, what’s the difference?

4. The so-called “kinetics control” in this manuscript was simply realized via the precursor weight ratio (MoO₃ 230-330mg, and S 6g), which forms a ratio from 3.8%~5.5%. Such a “kinetics control” seems crude. Based on such a description, the research community would be difficult to repeat these results

due to different growth systems, powder size of MoO₃, surface area of melted sulfur, pressure, and carrier gas flux. A precise description of the actual gaseous precursor flux, or partial pressure, should be more scientifically reasonable.

5. The unidirectional growth window (~4.5% ratio at 880 oC) seems extremely narrow, implying difficulty in reproducibility. How about the growth temperature window for the unidirectional epitaxy?

6. Line 72-78, "It is noteworthy that with the same trick, wafer-scale single-crystal MoS₂, as well as other 2D materials, have also been epitaxially grown on sapphire or other substrates²³⁻³²". The citation [23-32] should be checked. Ref. 29 did not report "wafer-scale single-crystals" and Ref. 30 finds translational grain boundaries and claimed only "unidirectional" growth. Some recent work of single-crystal MoSe₂ could be cited here.

7. Fig. 1d shows optical micrographs of MoS₂ at different growth stages, which show increased coverage and nucleation density. However, the crystal size did not increase, comparing the coalescence stage and the nucleation stage (See Fig. 1d). Please explain the reason.

8. The crystal size seems to be about 5~15 um, far smaller than the previous work (Nano Lett. 2020, 20, 7193–7199). Please explain the difference.

Reviewer #2 (Remarks to the Author):

This study demonstrated the epitaxial growth of 2'' single-crystal (SC) MoS₂ monolayers on c-plane sapphire by controlling the concentration ratio of the precursors. Different characterization methods have characterized and confirmed the film's alignment and seamless stitching. The epitaxial monolayer MoS₂ SC shows excellent uniformity and quality, as evidenced by the phonon circular dichroism, exciton valley polarization, high room-temperature mobility, and on/off ratio. The results are interesting. However, the paper requires a major revision, as suggested below, before further consideration.

1- The novelty of the present work with the recent studies in the field should be highlighted. Many literature have reported the growth of SC TMDs in the last few years. Although the final result is the same, different techniques have been used. Most of them could have the same mechanism. Nevertheless, they illustrated it from different perspectives due to the complicated growth mechanism. Most of these results are not reproducible due to the complicated factors of the growth. Considering all the growth factors, a general mechanism is needed in this stage to connect all the dots in the literature and explain the mechanism of SC epitaxial growth.

2- A very recent study (Nat. Nanotech. (2023): 1-6) has illustrated the effect of the off-cut angle of the c-

plane sapphire on the orientation of MoS₂, which partially explain the reason for achieving the SC growth on this kind of substrate. However, the growth conditions was not explained clearly in the Nat. Nanotech. paper. In the present study, the author should investigate the role of growth kinetics and relate that to other growth factors, like the structure of the steps, the substrate surface termination, and/or the seed composition.

3- The author highlighted the high quality of the film in the present work, but the reason for this high-quality grown film compared with the recent studies was not explained.

4- The system used in this study has three heating zones, making it easier to conduct a systematic study to understand and relate the critical factors to control the growth. The ratio of MoO₃/S could affect many things (the seed composition, the surface structure of the substrate, the structure of the step edges, etc.), yet the author did not investigate any of them. How does that affect the quality of the grown film?

5- The mobility was measured at high temperature, and the device was h-BN encapsulated, which can be considered the reason for the high mobility. FET measurements at room temperature are needed to compare with the literature.

Reviewer #3 (Remarks to the Author):

The authors present an approach to synthesize single-crystal MoS₂ on c-plane sapphire with an M-axis miscut through kinetic control. They offer sufficient evidence to substantiate the single crystallinity using optical microscopy, HADDF-STEM, LEED, and SHG. Additionally, they showcase MoS₂ transistors with record-high electron mobility. However, this work lacks a thorough understanding of the kinetic mechanism behind single-crystal MoS₂ growth. The methodology also lacks detailed procedures, creating challenges for readers attempting to replicate the results. Moreover, the explanations provided on the MoS₂ transistor include misleading information. Therefore, I would suggest this work should have a major revision to answer the following issue before making final decisions.

1. In line 79, the authors attempt to address a pivotal query: can MoS₂ growth be controlled by purely kinetic growth? Is this claim valid for pure c-plane sapphires without an M-axis miscut and many other substrates, such as different mistcut or surface plane sapphires?

2. Referring to a previous study (Nat Commun 14, 592 (2023)), it was posited that immature steps serve as nucleation sites for unidirectional growth. Based on this, might the S/Mo ratio be the crucial determinant controlling atomic step formation (t_s) that aligns with the growth time (t_g) as proposed by the authors? If not, please provide a comprehensive understanding of the kinetic growth mechanism by computational approaches, such as DFT, MD, or computational fluid dynamics.

3. According to references (Chem. Mater. 2014, 26, 22, 6371–6379, Nano Research volume 10, pages 255–262 (2017)), the S/Mo ratio typically dictates the shape evolution of MoS₂ crystals at varying furnace positions. Kindly provide optical microscopy images of CVD MoS₂ at different S/Mo ratios and locations, clarifying if the kinetic control mechanism remains consistent across these observations.
4. Regarding the growth procedure, conventional methods using sulfur and MoO₃ powders were employed. However, the positioning of S and Mo and the size of the openings for the sulfur and MoO₃ crucibles significantly influence the growth. It would be helpful to include photographs of the CVD setup and provide more experimental details for readers.
5. With the traditional S/MoO₃ growth, the quartz tube often develops a coating of Mo/MoO₃, which might be released during growth. Please compare results between brand-new quartz tubes and previously used ones.
6. In line 288, the authors assert that they achieved record-high mobility of 140 cm²V⁻¹S⁻¹ on a transistor with dimensions of 1 μm/5 μm. This value appears consistent with the short-channel device. The authors list mobility of 140 cm²V⁻¹S⁻¹ only for the short channel devices. Please confirm these figures and elucidate why the mobility of the long-channel matches that of the short-channel devices.
7. Please indicate the thickness of the hBN layers (both top and bottom) for the hBN-encapsulated devices.
8. As the transistor devices utilize dual gates for both long-channel hBN-encapsulated devices and short-channel HfO₂ devices, please measure the gate capacitances for both devices.

Point-by-point response to the Reviewers' comments

Reviewer #1 (Remarks to the Author):

The manuscript “Epitaxy of wafer-scale single-crystal MoS₂ monolayer via kinetic control” reported the unidirectional growth of MoS₂ and 2-inch single crystals on the general *c*-plane sapphire substrate via controlling the growth kinetics. This is an important research topic and attracts wide attention. The reviewer finds the results interesting but lacks a systematic study of the growth mechanism to support the innovation. Regarding the unidirectional MoS₂ growth, some important work in this field has raised their representative viewpoints, such as surface steps (<https://doi.org/10.1038/s41586-022-04523-5>), surface symmetry (<https://doi.org/10.1038/s41565-023-01445-9>), and surface chemistry (<https://doi.org/10.1038/s41565-023-01456-6>). The authors proposed that the kinetic control, i.e. a MoO₃/S ratio, plays a key role, which is possibly reasonable but lacks strict proof and deep-in discussion on the mechanisms. Following are some comments that the reviewer suggests the authors consider.

Response 1:

We sincerely thank the Reviewer for the positive evaluation on our work “*This is an important research topic and attracts wide attention*”. We also appreciate the Reviewer’s insightful and constructive comments for improvement.

We agree with the Reviewer that regarding the unidirectional MoS₂ growth, several previous studies have raised their representative viewpoints. We thank the Reviewer for pointing out these important works which have been cited in the revised manuscript.

To understand the underlying growth mechanism, we have performed additional characterizations following the Reviewer’s kind comments. Figures R1a and R1b show, respectively, the cross-sectional high-angle annular dark-field scanning transmission electron microscopy (HAADF-STEM) of the as-grown single-crystal MoS₂ along the $\langle 10\bar{1}0 \rangle$ (Fig. R1a) and $\langle 11\bar{2}0 \rangle$ directions of *c*-plane sapphire substrate (Fig. R1b). Importantly, we find that the *c*-plane sapphire surface is reconstructed to form an atomically thin buffer layer below the as-grown MoS₂. This buffer layer is the key, we believe, to facilitate the unidirectional epitaxy of the monolayer MoS₂. According to our control experiments, the buffer layer formation depends highly on the S/MoO₃ precursor ratio. For example, we have tried to anneal the *c*-plane sapphire in the atmosphere of pure S or MoO₃ for the following MoS₂ growth but we did not achieve the unidirectional growth. Under a particular ratio of precursors S/MoO₃, a specific buffer layer is formed, enabling the unidirectional domain alignment in the nucleation stage. It is noteworthy that similar results have recently been reported for MoS₂ epitaxy on β -Ga₂O₃ (001) that a particular S/MoO₃ precursor ratio results in the formation of a specific buffer layer and thus the unidirectional growth [e.g., *ACS Nano* 17, 10010 (2023)].

Figure R1. MoS₂ growth on α -Al₂O₃ (0001) with a buffer layer. **a,b**, Cross-sectional HAADF-STEM images of a MoS₂ grown on the α -Al₂O₃ (0001) substrate along the α -Al₂O₃ $\langle 10\bar{1}0 \rangle$ direction (**a**) and the α -Al₂O₃ $\langle 11\bar{2}0 \rangle$ direction (**b**).

To further confirm the effect of the buffer layer and its role on the unidirectional domain growth, we remove the as-grown unidirectional MoS₂ from *c*-plane sapphire substrate by water-assisted technique, as illustrated in Fig. R2a [please refer to: *Nat. Commun.* **11**, 2153 (2020); *ACS Nano* **11**, 12001 (2017)]. Note that such removal process utilizes the water intercalation and would not destroy the buffer layer. Then the sapphire substrates with buffer layer are used to re-grow MoS₂ under a condition where two antiparallel domains occur on unused new sapphire substrates. Note that we also put fresh sapphire substrates in the growth chamber at the same time for control samples. The results are shown in Fig. R2. On fresh sapphire substrates, two antiparallel domains appear simultaneously (Figs. R2c and R2d). By contrast, unidirectional MoS₂ domains are reproduced on the sapphire substrates with buffer layers (Figs. R2a and R2b). This result strongly demonstrates the key role of buffer layer on the unidirectional domain alignment. It is also worth noting that although an atomically thin buffer layer is observed via HAADF-STEM, it is still difficult to infer its exact structure. Further in-depth theoretical and experimental studies are required, while beyond the scope of this work.

To address this comment, we have added the above discussion into the revised manuscript (pages 4 and 5, lines 147-167), added Fig. R1 into Fig. 1 as Fig. 1d in the revised main text (page 4), and added Fig. R2 into Supporting Information as Supplementary Fig. 11 (page 8). In addition, considering that the growth mechanism should be the formation of a specific buffer layer by controlling the S/MoO₃ precursor ratio. In the revised manuscript, we have changed the phrase “kinetic control” to “buffer layer control” or “precursor ratio control”. For example, the title of our manuscript has been revised to “Epitaxy of wafer-scale single-crystal MoS₂ monolayer via buffer layer control”.

Figure R2. (a) Schematic illustration of removal and regrowth of unidirectional MoS₂. The as-grown unidirectional MoS₂ is removed from *c*-plane sapphire substrate by water-assisted technique, which would perfectly maintain the formed buffer layer. (b) Optical micrographs of the regrown MoS₂ with unidirectional domain alignment. (c,d) Illustration and optical micrographs of MoS₂ domains grown on an fresh sapphire substrate, showing two antiparallel domains.

1. The authors emphasize the kinetics control in the MoS₂ growth, but the reviewer does not find any detailed information/discussion about the kinetics, except simply mentioning the MoO₃/S ratio. What is the kinetics when we talk about it? How does the MoO₃/S ratio affect the growth kinetics? Why does the MoO₃/S ratio affect the domain alignment? Do the different MoO₃/S ratios produce different species contributing to the nucleation and growth, or produce different surface termination that affects the alignment?

Response 2:

We thank the Reviewer for the constructive comments. By performing cross-sectional HAADF-STEM characterizations and re-growing unidirectional MoS₂ on the *c*-plane sapphire with buffer layer, we infer that the unidirectional domain alignment should be determined by the specific buffer layer within the substrate-epilayer gap, which is formed under a particular ratio of precursors S/MoO₃ (please refer to our above response 1 for more details).

2. As claimed in a very recently published work in Nature Nanotechnology (<https://doi.org/10.1038/s41565-023-01445-9>), the bi-steps would produce unidirectional alignment due to surface symmetry, independently from the step direction. The authors should make a clear demonstration different from that work. For

this purpose, the substrate characterizations about the step height, and orientation should be supplemented:

- 1) XRD to check the miscut direction and miscut angle of the substrate (with unidirectional MoS₂ on it),
- 2) AFM characterization of surface step height (H) and terrace width (W), which should accordance with the XRD data ($\tan\theta=H/W$). Only mono-, non-M-steps are proved, the claimed unidirectional growth on the “general” substrate could be convincing.

Response 3:

Thanks for these kind suggestions. We agree with the Reviewer that in the work of *Nat. Nanotechnol.* (doi:10.1038/s41565-023-01445-9), the bi-steps and thus single-type surface symmetry are formed by annealing at high temperatures, which enable the unidirectional alignment. However, in our studies, the used c -plane sapphire substrates are the general substrates with mono-steps, rather than bi-steps. Please allow us to show it in detail.

1) Figure R3a shows the XRD rocking curve of our c -plane sapphire substrates with unidirectional MoS₂ on it, demonstrating a major miscut angle towards M axis of 0.135° , which agrees well with the parameters provided by the supplier of HeFei crystal Technical Material Co., Ltd. (c -plane sapphire with a major miscut angle towards M axis of $\sim 0.2 \pm 0.1^\circ$).

2) Figures R3b and R3c shows the AFM image of our c -plane sapphire substrates, indicating mono-step height H of ~ 0.16 nm and terrace width W of ~ 66.5 nm. The ratio between step height H and terrace width W , i.e., $H/W = 0.16\text{nm}/66.5\text{nm} = 0.0024$, accordance with the XRD data ($\tan 0.135^\circ = 0.002356$). Our measured results confirm that the sapphire substrates we used are the general substrates with mono-steps, in contrast to the bi-step substrates reported in the *Nat. Nanotechnol.* (doi:10.1038/s41565-023-01445-9).

3) Our c -plane sapphire substrates with a major miscut angle towards M axis of $\sim 0.2^\circ$ have mono-step are quite reasonable because our substrates are annealing at 980°C , much lower than the required temperature ($>1400^\circ\text{C}$) to form bi-steps [*Nat. Nanotechnol.* doi:10.1038/s41565-023-01445-9 (2023)].

To address this comment, we have added above discussion and Figure R3 into Supporting Information as Supplementary Note 6 (pages 6-7, lines 78-98).

Figure R3. (a) XRD rocking curve of *c*-plane sapphire substrates with unidirectional MoS₂ on it, demonstrating a major miscut angle towards M axis of 0.135°. (b,c) AFM characterization of sapphire substrates, indicating surface step height of ~0.16 nm and terrace width of ~66.5 nm.

3. A recent work published in Nature Communications (<https://doi.org/10.1038/s41467-023-36286-6>) from the same united groups claimed universal epitaxy of single crystal TMDs on sapphire by controlling the simultaneous formation of grain nuclei and substrate steps, where step direction is not a requisite. Does the present work follow the same mechanism as that one? If not, what's the difference?

Response 4:

We thank the Reviewer for pointing out our previous work of *Nat. Commun.* **14**, 592 (2023). In our prior work of *Nat. Commun.* **14**, 592 (2023), single crystal TMDs are epitaxially grown on sapphire by controlling the simultaneous formation of grain nuclei and substrate steps. We proposed that the growth follows a dual-coupling mechanism, that is, the TMD epitaxy is guided by both the terraces and the step edges of the substrate. By contrast, in the present work, the substrates are pre-annealed at 980°C for 4 hours to form atomically smooth surfaces with a typical terrace width of ~66 nm and height of ~0.2 nm (Fig. R4) prior to the growth. By precisely controlling the S/MoO₃ precursor ratio, we achieve the unidirectional domain alignment assisted from the formation of an interfacial buffer layer between the substrate and epilayer (please refer to our above response 1 for more details).

Figure R4. The AFM characterization of the universal sapphire before annealed (a) and after annealed at 980°C for 4h (b).

4. The so-called “kinetics control” in this manuscript was simply realized via the precursor weight ratio (MoO₃ 230-330mg, and S 6g), which forms a ratio from 3.8%~5.5%. Such a “kinetics control” seems crude. Based on such a description, the research community would be difficult to repeat these results due to different growth systems, powder size of MoO₃, surface area of melted sulfur, pressure, and carrier gas flux. A precise description of the actual gaseous precursor flux, or partial pressure, should be more scientifically reasonable.

Response 5:

We fully agree and thank the Reviewer for the kind suggestions. Figure R5 (Fig. R6) shows the schematic diagram (photographs) of our chemical vapor deposition (CVD) setup for the epitaxy of monolayer MoS₂. For a typical growth, the center mini-tube is loaded with sulfur (Alfa Aesar, 99.9%, 6 g) and flowed with 40 sccm Ar. Note that to effectively load sulfur, a large rectangular chamber is designed ~5 cm from the end of the quartz tube. The surface area of melted sulfur determined by the rectangular chamber is around 25cm². For the outside six mini-tubes, every-other-tube is loaded with MoO₃ (Alfa Aesar, 99.999%) and flowed with Ar/O₂ (40/0.5 sccm); and the other three empty tubes are also flowed with Ar/O₂ (40/0.5 sccm). In addition, to avoid the powder being blown away by the carrier gas, we use MoO₃ thin flack (the thickness is ~0.5 mm) which is obtained by pressing the MoO₃ powder (Alfa Aesar, 99.999%) with a hydraulic press. All the quartz tubes have an inner diameter of ~1 cm. The distance between MoO₃ and (substrate) sulfur is ~32cm-40cm (~17 cm). During the growth, sulfur, MoO₃ and sapphire substrate are placed at first, second, and third temperature zones, respectively. The temperatures for the sulfur, MoO₃, and sapphire substrate are 120, 560, and 880 °C, respectively. The pressure is kept at about 1 torr during the growth process.

To address this comment, we have added above growth details into the EXPERIMENTAL SECTION of revised manuscript (page 10, lines 342-355). In addition, Figures R5 and R6 have been added into Supporting Information as Supplementary Fig. 24 and Fig. 25 (page 16).

Figure R5. Schematic diagram of the multisource CVD setup.

Figure R6. Photographs of our three-temperature-zone CVD setup and the location of the solid-state sources and the substrate.

5. The unidirectional growth window ($\sim 4.5\%$ ratio at 880°C) seems extremely narrow, implying difficulty in reproducibility. How about the growth temperature window for the unidirectional epitaxy?

Response 6:

We appreciate the Reviewer's comment. In addition to the unidirectional epitaxy at 880°C , we also achieve the unidirectional growth at 850°C (Fig. R7). However, at different growth temperatures, the corresponding S/MoO₃ precursor ratio is also different. For example, Figure R8 shows the degree of unidirectional alignment against the MoO₃/S precursor ratios at a growth temperature of 850°C . Clearly, the best MoO₃/S precursor ratio that show $\sim 100\%$ unidirectional alignment is $\sim 2.1\%$. This is contrast to the best MoO₃/S precursor ratio of $\sim 4.5\%$ at a growth temperature of 880°C .

To address this comment, we have added above discussion and Figs. R7/R8 into Supporting Information as Supplementary Note 18 (pages 17-18, lines 280-293).

Figure R7. Unidirectional growth of MoS₂ at different growth temperatures and different growth conditions.

Figure R8. Proportional changes of the single oriented MoS₂ domains grown by varying the MoO₃/S ratio at a growth temperature of 850°C.

6. Line 72-78, “It is noteworthy that with the same trick, wafer-scale single-crystal MoS₂, as well as other 2D materials, have also been epitaxially grown on sapphire or other substrates²³⁻³²”. The citation [23-32] should be checked.

Ref. 29 did not report “wafer-scale single-crystals” and Ref. 30 finds translational grain boundaries and claimed only “unidirectional” growth. Some recent work of single-crystal MoSe₂ could be cited here.

Response 7:

We fully agree with the Reviewer that Ref. 29 did not report “wafer-scale single-crystals” and Ref. 30 finds translational grain boundaries and claimed only “unidirectional” growth. In the revised the manuscript, we have removed these two referees mentioned by the Reviewer. In addition, following the Reviewer’s suggestion, we have added some new referees in the revised manuscript, such as recent works of single-crystal MoSe₂ [*Natl. Sci. Open* 2, 20220055 (2023); *Nat. Sci.* 3, 20220059 (2023)] and these recent works published during the review period of our work [*Nat. Nanotechnol.* doi:10.1038/s41565-023-01445-9 (2023); *Nat. Nanotechnol.*, doi:10.1038/s41565-023-01456-6 (2023)].

7. Fig. 1d shows optical micrographs of MoS₂ at different growth stages, which show increased coverage and nucleation density. However, the crystal size did not increase, comparing the coalescence stage and the nucleation stage (See Fig. 1d). Please explain the reason.

Response 8:

Sorry for the confusion. In fact, for the left panel of Fig. 1d in the main text, it is the optical image with growth time of ~ 15 min, rather than at the nucleation stage. In other words, the observed MoS₂ domains have grown nicely. Figure R9 shows more optical micrographs of unidirectional MoS₂ grown on *c*-plane sapphire at different times. It can be seen clearly that the crystal size at growth time of ~ 15 min increases significantly compared to that at growth time of ~ 5 min.

In the revised manuscript, the optical micrographs of MoS₂ at different growth stages have been revised as Fig. R10 for clearer view.

Figure R9. Optical images of unidirectional MoS₂ for different grown times.

Figure R10. Optical micrographs of unidirectional MoS₂ grown on *c*-plane sapphire at different times

8. The crystal size seems to be about 5~15 μm , far smaller than the previous work (*Nano Lett.* 2020, 20, 7193–7199). Please explain the difference.

Response 9:

We thank the Reviewer for pointing out our previous work of *Nano Lett.* 20, 7193 (2020). We agree with the Reviewer that the crystal size in the present work is about 15 μm , smaller than our previous work of *Nano Lett.* 20, 7193 (2020) where the crystal size can be larger than 100 μm . The much larger crystal size in our previous work of *Nano Lett.* 20, 7193 (2020) is due to a larger oxygen flow (10 sccm) we used. According our previous results [*Nano Lett.* 20, 7193 (2020); *J. Am. Chem. Soc.* 137, 15632 (2015)], a large oxygen flow facilitates a great crystal size by decreasing the nucleation density and simultaneously increasing the growth rate. In the present work, to realize growth of single-crystal MoS₂ monolayer, we find that a small oxygen flow is required (3 sccm), resulting in smaller crystal size.

We thank the Reviewer for his/her very valuable reviewing efforts. We hope we have convincingly addressed all the comments raised by the Reviewer that our revised manuscript meets the criteria for publication in *Nature Communications*.

Reviewer #2 (Remarks to the Author):

This study demonstrated the epitaxial growth of 2-inch single-crystal (SC) MoS₂ monolayers on c-plane sapphire by controlling the concentration ratio of the precursors. Different characterization methods have characterized and confirmed the film's alignment and seamless stitching. The epitaxial monolayer MoS₂ SC shows excellent uniformity and quality, as evidenced by the phonon circular dichroism, exciton valley polarization, high room-temperature mobility, and on/off ratio. The results are interesting. However, the paper requires a major revision, as suggested below, before further consideration.

Response 10:

We sincerely thank the Reviewer for the positive evaluation on our work “*The results are interesting*”. We also appreciate the Reviewer’s insightful and constructive comments for improvement. Below we address the Reviewer’s comments point by point.

1- The novelty of the present work with the recent studies in the field should be highlighted. Many literature have reported the growth of SC TMDs in the last few years. Although the final result is the same, different techniques have been used. Most of them could have the same mechanism. Nevertheless, they illustrated it from different perspectives due to the complicated growth mechanism. Most of these results are not reproducible due to the complicated factors of the growth. Considering all the growth factors, a general mechanism is needed in this stage to connect all the dots in the literature and explain the mechanism of SC epitaxial growth.

Response 11:

We appreciate the Reviewer’s comment. We fully agree with the Reviewer that some works have reported the growth of SC TMDs on sapphire substrates in the recent years, such as *Nat. Nanotechnol.* **16**, 1201 (2021), *Nat. Nanotechnol.* **17**, 33 (2022), *Nat. Nanotechnol.*, doi:10.1038/s41565-023-01445-9 (2023) and *Nat. Nanotechnol.*, doi:10.1038/s41565-023-01456-6 (2023). However, these works rely heavily on the surface step engineering (e.g., controlling the surface step orientation and height), which typically requires the specially-designed substrates such as deliberately engineered off-cut angles or annealing at harsh temperatures. Here, we realize the epitaxial growth of wafer-scale SC MoS₂ monolayers on industry-compatible of general c-plane sapphire substrates by simply controlling the S/MoO₃ precursor ratio, offering a new insight into the synthesis of wafer-scale high-quality 2D semiconductors and thus showing high novelty.

Regarding the general mechanism, a universal symmetry framework has been theoretically established to guide the growth of wafer-scale SC 2D materials, i.e., the symmetry group of the substrate should be a subgroup of 2D material [*Nat. Commun.* **11**, 5862 (2020)]. In the light of such general guideline, c-plane sapphire with C_{3v} symmetry offers a possible industry-compatible substrate for the epitaxial growth of wafer-scale SC MoS₂ with point group D_{3h} (C_{3v} plus a mirror-reflection symmetry σ_h).

However, the adsorption energy of the most preferred domain orientation (i.e., 30°) is just slightly lower than that at 90° configuration. This usually results in two antiparallel grains and twin boundaries, as in the earlier results [*ACS Nano* **11**, 12001 (2017); *ACS Nano* **11**, 9215 (2017)]. Consequently, the key strategies to achieve the growth of wafer-scale SC MoS₂ is to enlarge the binding energy difference between the most preferred domain orientation and its antiparallel domain. For previously reported wafer-scale growth of SC TMDs via surface step engineering, it does follow the general guideline where the sapphire step edge-TMDs interaction breaks the symmetry of the antiparallel orientations. In our work, we enlarge the binding energy difference between the most preferred domain orientation and its antiparallel domain to realize wafer-scale SC MoS₂ by precisely controlling the S/MoO₃ precursor ratio.

To address this comment, we have revised the introduction part of our manuscript to highlight the novelty of our work, and added a section (i.e., Supplementary Note 19) in the Supporting Information to discuss the general mechanism of the SC TMD growth (page 18, lines 295-320).

2- A very recent study (*Nat. Nanotech.* (2023): 1-6) has illustrated the effect of the off-cut angle of the *c*-plane sapphire on the orientation of MoS₂, which partially explain the reason for achieving the SC growth on this kind of substrate. However, the growth conditions was not explained clearly in the *Nat. Nanotech.* Paper. In the present study, the author should investigate the role of growth kinetics and relate that to other growth factors, like the structure of the steps, the substrate surface termination, and/or the seed composition.

Response 12:

Thanks for the kind suggestions. If we are right, the *Nat. Nanotech.* paper mentioned by the Reviewer should be *Nat. Nanotechnol.* doi:10.1038/s41565-023-01445-9 (2023). In this work of *Nat. Nanotechnol.* doi:10.1038/s41565-023-01445-9 (2023), the bi-steps and thus single-type surface symmetry are formed by introducing a larger miscut angle together with high temperature annealing, which is believed to enable the SC growth. By contrast, our work investigates the role of the S/MoO₃ precursor ratio and realizes the SC growth on the general *c*-plane sapphire substrates with mono-step, rather than bi-steps.

We fully agree with the Reviewer that we should relate the S/MoO₃ precursor ratio control to other growth factors. Following the Reviewer's kind comments, we have performed additional characterizations. Figures R11a and R11b show, respectively, the cross-sectional high-angle annular dark-field scanning transmission electron microscopy (HAADF-STEM) of the as-grown single-crystal MoS₂ along the $\langle 10\bar{1}0 \rangle$ (Fig. R11a) and $\langle 11\bar{2}0 \rangle$ directions of *c*-plane sapphire substrate (Fig. R11b). Importantly, we find that the *c*-plane sapphire surface is reconstructed to form an atomically thin buffer layer below the as-grown MoS₂. This buffer layer is the key, we believe, to facilitate the unidirectional epitaxy of the monolayer MoS₂. According to our control experiments, the buffer layer formation depends highly on the S/MoO₃

precursor ratio. For example, we have tried to anneal the *c*-plane sapphire in the atmosphere of pure S or MoO₃ for the following MoS₂ growth but we did not achieve the unidirectional growth. Under a particular ratio of precursors S/MoO₃, a specific buffer layer is formed, enabling the unidirectional domain alignment in the nucleation stage. It is noteworthy that similar results have recently been reported for MoS₂ epitaxy on β -Ga₂O₃ (001) that a particular S/MoO₃ precursor ratio results in the formation of a specific buffer layer and thus the unidirectional growth [e.g., *ACS Nano* 17, 10010 (2023)].

Figure R11. MoS₂ growth on α -Al₂O₃ (0001) with a buffer layer. **a,b**, Cross-sectional HAADF-STEM images of a MoS₂ grown on the α -Al₂O₃ (0001) substrate along the α -Al₂O₃ $\langle 10\bar{1}0 \rangle$ direction (**a**) and the α -Al₂O₃ $\langle 11\bar{2}0 \rangle$ direction (**b**).

To further confirm the effect of the buffer layer and its role on the unidirectional domain growth, we remove the as-grown unidirectional MoS₂ from *c*-plane sapphire substrate by water-assisted technique, as illustrated in Fig. R12a [please refer to: *Nat. Commun.* 11, 2153 (2020); *ACS Nano* 11, 12001 (2017)]. Note that such removal process utilizes the water intercalation and would not destroy the buffer layer. Then the sapphire substrates with buffer layer are used to re-grow MoS₂ under a condition where two antiparallel domains occur on unused new sapphire substrates. Note that we also put fresh sapphire substrates in the growth chamber at the same time for control samples. The results are shown in Fig. R12. On fresh sapphire substrates, two antiparallel domains appear simultaneously (Figs. R12c and R12d). By contrast, unidirectional MoS₂ domains are reproduced on the sapphire substrates with buffer layers (Figs. R12a and R12b). This result strongly demonstrates the key role of buffer layer on the unidirectional domain alignment. It is also worth noting that although an atomically thin buffer layer is observed via HAADF-STEM, it is still difficult to infer its exact structure. Further in-depth theoretical and experimental studies are required but beyond the scope of this work.

To address this comment, we have added the above discussion into the revised manuscript (pages 4 and 5, lines 147-167), added Fig. R11 into Fig. 1 as Fig. 1d in the revised main text (page 4), and added Fig. R12 into Supporting Information as Supplementary Fig. 11 (page 8). In addition, considering that the growth mechanism

should be the formation of a specific buffer layer by controlling the S/MoO₃ precursor ratio. In the revised manuscript, we have changed the phrase “kinetic control” to “buffer layer control” or “precursor ratio control”. For example, the title of our manuscript has been revised to “Epitaxy of wafer-scale single-crystal MoS₂ monolayer via buffer layer control”.

Figure R12. (a) Schematic illustration of removal and regrowth of unidirectional MoS₂. The as-grown unidirectional MoS₂ is removed from *c*-plane sapphire substrate by water-assisted technique, which would perfectly maintain the formed buffer layer. (b) Optical micrographs of the regrown MoS₂ with unidirectional domain alignment. (c,d) Illustration and optical micrographs of MoS₂ domains grown on an fresh sapphire substrate, showing two antiparallel domains.

3- The author highlighted the high quality of the film in the present work, but the reason for this high-quality grown film compared with the recent studies was not explained.

Response 13:

We appreciate the Reviewer’s comment. The high-quality of our grown SC MoS₂ film can be understood from several aspects. First, we keep a sulfur-rich condition during the growth, facilitating achieving a low density of sulfur vacancies and thus the high quality of the as-grown SC MoS₂. To confirm this, we have performed a statistical analysis of the atomic-resolution STEM images and found that the density of sulfur vacancies in our monolayer MoS₂ is $\sim 5.2 \times 10^{12} \text{cm}^{-2}$ (Fig. R13), which is an order of magnitude lower than that of the previously reported exfoliated flakes [e.g., *Nat. Commun.* **6**, 6293 (2015)]. Second, our chemical vapor deposition (CVD) setup has a unique multisource design (please see Fig. R14 for the schematic diagram of our multisource CVD setup), which facilitates the homogeneous cross-sectional source supply and thus high quality. Third, our fabricated monolayer MoS₂ devices have high field effect mobilities at room temperature partially benefiting from the *h*-BN

capsulation (to achieve better interface quality) and the dual-gate modulation (to achieve higher carrier densities). Besides, our SC MoS₂ films are stitched from unidirectional domains on mono-step sapphire surfaces and the nucleation and growth of domains don't rely on the surface steps, thus we believe that the high-quality of as-grown films would also benefit from such good stitching.

To address this comment, we have added above discussion into the revised manuscript (page 7, line 257-265) and added Fig. R14 into Supporting Information as Supplementary Fig. 24 (page 16).

Figure R13. The atomic-resolution STEM images of MoS₂ samples.

Figure R14. Schematic diagram of the multisource CVD setup.

4- The system used in this study has three heating zones, making it easier to conduct a systematic study to understand and relate the critical factors to control the growth. The ratio of MoO₃/S could affect many things (the seed composition, the surface structure

of the substrate, the structure of the step edges, etc.), yet the author did not investigate any of them. How does that affect the quality of the grown film?

Response 14:

We agree with the Reviewer that the ratio of MoO₃/S could affect many things. By performing cross-sectional HAADF-STEM characterizations and re-growing unidirectional MoS₂ on the *c*-plane sapphire with unidirectional MoS₂ removed, we can infer that a particular ratio of precursors S/MoO₃ would result in the formation of a specific buffer layer between the substrate and MoS₂ epilayer. Such a specific buffer layer plays a key role for the unidirectional domain alignment (please refer to our above response 12 for more details). We believe that such an atomically thin buffer layer has essentially no effect on the quality of the grown film, as confirmed by the high quality of our epitaxial monolayer MoS₂ single crystals.

Further, we have also investigated the effect of substrate temperature. In addition to the unidirectional epitaxy at 880°C, we also achieve the unidirectional growth at 850°C (Fig. R15). However, at different growth temperatures, the corresponding S/MoO₃ precursor ratio is also different. For example, Fig. R16 shows the degree of unidirectional alignment against the MoO₃/S precursor ratios at a growth temperature of 850°C. Clearly, the best MoO₃/S precursor ratio that show ~100% unidirectional alignment is ~2.1%. This is contrast to the best MoO₃/S precursor ratio of ~4.5% at a growth temperature of 880°C (Fig. 1c in the main text).

To address this comment, we have added above discussion and Figs. R15/R16 into Supporting Information as Supplementary Note 18 (pages 17-18, lines 280-293).

Figure R15. Unidirectional growth of MoS₂ at different growth temperatures.

Figure R16. Proportional changes of the single oriented MoS₂ domains grown by varying the MoO₃/S ratio at a growth temperature of 850°C.

5- The mobility was measured at high temperature, and the device was h-BN

encapsulated, which can be considered the reason for the high mobility. FET measurements at room temperature are needed to compare with the literature.

Response 15:

Sorry for the confusion. In fact, the mobilities of all our devices are measured at room temperature, rather than high temperature. To avoid confusion, we have changed the sentence “the extracted field-effect mobility ...can reach $\sim 140 \text{ cm}^2\text{s}^{-1}\text{V}^{-1}$ ” to “the extracted field-effect mobility at room-temperature...can reach $\sim 140 \text{ cm}^2\text{s}^{-1}\text{V}^{-1}$ ” (page 9, lines 297-299). In addition, we also add a sentence “Electrical measurements were carried out with an Agilent B1500 semiconductor parameter analyzer in a four-probe vacuum station with a bass pressure of $\sim 10^{-6}$ mbar at room temperature” into the EXPERIMENTAL SECTION (page 12, lines 416-417).

We agree with the Reviewer that *h*-BN encapsulation can be considered the reason for the high mobility. The extracted room-temperature mobility of $\sim 140 \text{ cm}^2\text{s}^{-1}\text{V}^{-1}$ in our devices is larger than the results based on polycrystalline films and competitive to record-high value of the exfoliated flakes [*Nature* **520**, 656 (2015); *Nature* **568**, 70 (2019); *Nat. Nanotechnol.* **16**, 1201 (2021)].

We thank the Reviewer for his/her very valuable reviewing efforts. We hope we have convincingly addressed all the comments raised by the Reviewer that our revised manuscript meets the criteria for publication in *Nature Communications*.

Reviewer #3 (Remarks to the Author):

The authors present an approach to synthesize single-crystal MoS₂ on c-plane sapphire with an M-axis miscut through kinetic control. They offer sufficient evidence to substantiate the single crystallinity using optical microscopy, HADDF-STEM, LEED, and SHG. Additionally, they showcase MoS₂ transistors with record-high electron mobility. However, this work lacks a thorough understanding of the kinetic mechanism behind single-crystal MoS₂ growth. The methodology also lacks detailed procedures, creating challenges for readers attempting to replicate the results.

Moreover, the explanations provided on the MoS₂ transistor include misleading information. Therefore, I would suggest this work should have a major revision to answer the following issue before making final decisions.

Response 16:

We sincerely thank the Reviewer for the positive evaluation on our work “*They offer sufficient evidence to substantiate the single crystallinity*”. We also appreciate the Reviewer’s insightful and constructive comments for improvement. Below we address the Reviewer’s comments point by point.

1. In line 79, the authors attempt to address a pivotal query: can MoS₂ growth be controlled by purely kinetic growth? Is this claim valid for pure c-plane sapphires without an M-axis miscut and many other substrates, such as different miscut or surface plane sapphires?

Response 17:

We appreciate the Reviewer’s insightful comment. In addition to c-plane sapphire substrates with a major miscut angle ($\sim 0.2^\circ$) towards M-axis in the main text, the unidirectional domain alignment has also been achieved by precisely controlling the S/MoO₃ precursor ratio on pure c-plane sapphires without an M-axis miscut (the degree of unidirectional alignment is $\sim 99.2\%$, Fig. R17a), c-plane sapphire substrates with different major miscut angles towards M-axis such as C/M- 0.5° (the degree of unidirectional alignment is $\sim 96.6\%$, Fig. R17b) and C/M- 3° (the degree of unidirectional alignment is $\sim 99.5\%$, Fig. R17c), c-plane sapphire substrates with a major miscut angle ($\sim 0.2^\circ$) towards other axes (the degree of unidirectional alignment is $\sim 98.7\%$ in Fig. R17d and the degree of unidirectional alignment is $\sim 96.4\%$ in Fig. R17e), and c-plane sapphire substrates with a major miscut angle ($\sim 0.2^\circ$) towards A-axis (the degree of unidirectional alignment is $\sim 98.1\%$, Fig. R17f). Therefore, we believe that our claim is valid for pure c-plane sapphires without an M-axis miscut and other substrates.

To address this comment, we have added a sentence “In addition to c-plane sapphire substrates with a major miscut angle ($\sim 0.2^\circ$) towards M-axis, the unidirectional domain alignment has also been achieved by precisely controlling the S/MoO₃ precursor ratio on pure c-plane sapphires, and c-plane sapphire substrates with different major miscut angles towards other axes (please see Supplementary Note 3), indicating the universality of our method.” in the revised manuscript (page 3, lines 111-115) and

added Fig. R17 and associated discussions into Supporting Information as Supplementary Note 3 (pages 3-4, lines 31-50).

Figure R17. Optical microscopy image of unidirectional MoS₂ domains on pure *c*-plane sapphires without an M-axis miscut (a), *c*-plane sapphire substrates with a 0.5° major miscut angles towards M-axis (b), a 3° major miscut angles towards M-axis (c), with a 0.2° major miscut angles towards other axes (d,e), and with a 0.2° major miscut angle towards A-axis (f).

2. Referring to a previous study (Nat Commun 14, 592 (2023)), it was posited that immature steps serve as nucleation sites for unidirectional growth. Based on this, might the S/Mo ratio be the crucial determinant controlling atomic step formation (ts) that aligns with the growth time (ts) as proposed by the authors? If not, please provide a comprehensive understanding of the kinetic growth mechanism by computational approaches, such as DFT, MD, or computational fluid dynamics.

Response 18:

We thank the Reviewer for pointing out our prior work of *Nat. Commun.* 14, 592 (2023). In the present work, the *c*-plane sapphire substrates are annealing at 980°C for 4 hours before the growth. In other words, atomically smooth surfaces with a typical terrace width of ~66 nm and step height of ~0.2 nm is formed before the growth (Fig. R18). This is in contrast to our prior work of *Nat. Commun.* 14, 592 (2023) that the formation of grain nuclei and substrate steps occurs simultaneously. Therefore, our present work is different from the prior work of *Nat. Commun.* 14, 592 (2023).

Figure R18. The AFM characterization of the universal sapphire before annealed (a) and after annealed at 980°C for 4h (b).

To understand the underlying growth mechanism, we have performed additional characterizations following the Reviewer's kind comments. Figures R19a and R19b show, respectively, the cross-sectional high-angle annular dark-field scanning transmission electron microscopy (HAADF-STEM) of the as-grown single-crystal MoS₂ along the $\langle 10\bar{1}0 \rangle$ (Fig. R19a) and $\langle 11\bar{2}0 \rangle$ directions of *c*-plane sapphire substrate (Fig. R19b). Importantly, we find that the *c*-plane sapphire surface is reconstructed to form an atomically thin buffer layer below the as-grown MoS₂. This buffer layer is the key, we believe, to facilitate the unidirectional epitaxy of the monolayer MoS₂. According to our control experiments, the buffer layer formation depends highly on the S/MoO₃ precursor ratio. For example, we have tried to anneal the *c*-plane sapphire in the atmosphere of pure S or MoO₃ for the following MoS₂ growth but we did not achieve the unidirectional growth. Under a particular ratio of precursors S/MoO₃, a specific buffer layer is formed, enabling the unidirectional domain alignment in the nucleation stage. It is noteworthy that similar results have recently been reported for MoS₂ epitaxy on β -Ga₂O₃ (001) that a particular S/MoO₃ precursor ratio results in the formation of a specific buffer layer and thus the unidirectional growth [e.g., *ACS Nano* 17, 10010 (2023)].

Figure R19. MoS₂ growth on α -Al₂O₃ (0001) with a buffer layer. **a,b**, Cross-sectional HAADF-STEM images of a MoS₂ grown on the α -Al₂O₃ (0001) substrate along the α -Al₂O₃ $\langle 10\bar{1}0 \rangle$ direction (a) and the α -Al₂O₃ $\langle 11\bar{2}0 \rangle$ direction (b).

To further confirm the effect of the buffer layer and its role on the unidirectional domain growth, we remove the as-grown unidirectional MoS₂ from *c*-plane sapphire substrate by water-assisted technique, as illustrated in Fig. R20a [please refer to: *Nat. Commun.* 11, 2153 (2020); *ACS Nano* 11, 12001 (2017)]. Note that such removal process utilizes the water intercalation and would not destroy the buffer layer. Then the sapphire substrates with buffer layer are used to re-grow MoS₂ under a condition where two antiparallel domains occur on unused new sapphire substrates. Note that we also put fresh sapphire substrates in the growth chamber at the same time for control samples. The results are shown in Fig. R20. On fresh sapphire substrates, two antiparallel

domains appear simultaneously (Figs. R20c and R20d). By contrast, unidirectional MoS₂ domains are reproduced on the sapphire substrates with buffer layers (Figs. R20a and R20b). This result strongly demonstrates the key role of buffer layer on the unidirectional domain alignment. It is also worth noting that although an atomically thin buffer layer is observed via HAADF-STEM, it is still difficult to infer its exact structure. Further in-depth theoretical and experimental studies are required, while beyond the scope of this work.

To address this comment, we have added the above discussion into the revised manuscript (pages 4 and 5, lines 147-167), added Fig. R19 into Fig. 1 as Fig. 1d in the revised main text (page 4), and added Fig. R20 into Supporting Information as Supplementary Figs. 11 (page 8). In addition, considering that the growth mechanism should be the formation of a specific buffer layer by controlling the S/MoO₃ precursor ratio. In the revised manuscript, we have changed the phrase “kinetic control” to “buffer layer control” or “precursor ratio control”. For example, the title of our manuscript has been revised to “Epitaxy of wafer-scale single-crystal MoS₂ monolayer via buffer layer control”.

Figure R20. (a) Schematic illustration of removal and regrowth of unidirectional MoS₂. The as-grown unidirectional MoS₂ is removed from *c*-plane sapphire substrate by water-assisted technique, which would perfectly maintain the formed buffer layer. (b) Optical micrographs of the regrown MoS₂ with unidirectional domain alignment. (c,d) Illustration and optical micrographs of MoS₂ domains grown on an fresh sapphire substrate, showing two antiparallel domains.

3. According to references (Chem. Mater. 2014, 26, 22, 6371–6379, Nano Research volume 10, pages255–262 (2017)), the S/Mo ratio typically dictates the shape evolution of MoS₂ crystals at varying furnace positions. Kindly provide optical microscopy

images of CVD MoS₂ at different S/Mo ratios and locations, clarifying if the kinetic control mechanism remains consistent across these observations.

Response 19:

We thank the Reviewer for referring these two nice works. In both the two papers mentioned by the Reviewer, the substrate is very close to the MoO₃ source, while the S source is ~18 cm from the substrate (Fig. R21a). Consequently, there would be an obvious concentration gradient of MoO₃ on the surface of substrates during the growth, while the S vapor gradient on the substrate can be negligible. As the local position of substrate moves away from the MoO₃ source, it can gradually change from a Mo-rich to a S-rich condition, leading to the shape evolution of MoS₂ crystals at varying furnace positions.

By contrast, for our home-designed three-temperature-zone chemical vapor deposition (CVD) setup, there is a long distance between the substrate and MoO₃ source (>32 cm), as well as S source (>49 cm), as shown in Fig. R21b. Therefore, the gradient of both MoO₃ and S sources is very small and the MoS₂ crystals would maintain the same shape. Figure R22 shows the optical microscopy images of CVD MoS₂ at different locations (~32-40 cm). Clearly, the MoS₂ crystals keep the triangular shape. Similar results have been observed in other papers where their CVD systems also have a long distance between the substrate and MoO₃ source, as well as S source [*ACS Nano* 11, 9215 (2017); *ACS Nano* 12, 10032 (2018)].

Furthermore, for all the MoO₃/S ratios we have investigated (ranging from ~3.83% to ~5.55%), the concentration of S source is always larger than that of MoO₃ source. Consequently, the shape of MoS₂ crystals would keep unchanged, as in our experimental results (Fig. R22).

To address this comment, we have added a sentence “It is noteworthy that for all the MoO₃/S ratios we have investigated (ranging from ~3.83% to ~5.55%), it always belongs to a S-rich condition. Consequently, the shape of MoS₂ crystals remains unchanged (please see Supplementary Note 4), in contrast prior work where the drastic changes in S/Mo ratio lead to the shape evolution^{37,38}” in the revised manuscript (page 3, lines 115-119) and added Fig. R21b/Fig. R22 into Supporting Information as Supplementary Fig. 24/Fig. 4 (page 16/5).

Ref. 37: *Chem. Mater.* 26, 6371–6379 (2014);

Ref. 38: *Nano Research* 10, 255–262 (2017).

a

[REDACTED]

b

Figure R21. (a) Schematic of the CVD system used in *Chem. Mater.* 26, 6371–6379 (2014). (b) Illustration of our home-designed three-temperature-zone CVD setup.

Figure R22. Representative optical images of MoS₂ crystals grown at different MoO₃/S (a)-(d) at several different positions on the substrate (1)-(4).

4. Regarding the growth procedure, conventional methods using sulfur and MoO₃

powders were employed. However, the positioning of S and Mo and the size of the openings for the sulfur and MoO₃ crucibles significantly influence the growth. It would be helpful to include photographs of the CVD setup and provide more experimental details for readers.

Response 20:

We thank the Reviewer for the kind suggestion. Figure R21b shows the schematic diagram of our specially designed multisource CVD setup. Figure R23 presents the photographs of the CVD setup and the location of the S/MoO₃ sources. For a typical growth, the center mini-tube is loaded with sulfur (Alfa Aesar, 99.9%, 6 g) and flowed with 40 sccm Ar. Note that to effectively load sulfur, a large rectangular chamber is designed ~5 cm from the end of the quartz tube. The surface area of melted sulfur determined by the rectangular chamber is around 25cm². For the outside six mini-tubes, every-other-tube is loaded with MoO₃ (Alfa Aesar, 99.999%) and flowed with Ar/O₂ (40/0.5 sccm); and the other three empty tubes are also flowed with Ar/O₂ (40/0.5 sccm). In addition, to avoid the powder being blown away by the carrier gas, we use MoO₃ thin flack (the thickness is ~0.5 mm) which is obtained by pressing the MoO₃ powder (Alfa Aesar, 99.999%) with a hydraulic press. All the quartz tubes have an inner diameter of ~1 cm. The distance between MoO₃ and (substrate) sulfur is ~32cm-40cm (~17 cm). During the growth, sulfur, MoO₃ and sapphire substrate are placed at first, second, and third temperature zones, respectively. The temperatures for the sulfur, MoO₃, and sapphire substrate are 120, 560, and 880 °C, respectively. The pressure is kept at about 1 torr during the growth process.

To address this comment, we have added above growth details into the EXPERIMENTAL SECTION of revised manuscript (page 10, lines 342-355). In addition, Figure R23 has been added into Supporting Information as Supplementary Fig. 25 (page 16).

Figure R23. Photographs of our three-temperature-zone CVD setup and the location of the solid-state sources and the substrate.

5. With the traditional S/MoO₃ growth, the quartz tube often develops a coating of Mo/MoO₃, which might be released during growth. Please compare results between brand-new quartz tubes and previously used ones.

Response 21:

We fully agree with the Reviewer that the quartz tube would develop a coating of Mo/MoO₃ after the growth, as shown in Fig. R24. To eliminate this effect, we will anneal the used quartz tubes is placed in an air atmosphere at a high temperature (>1000°C) for 1h after the growth, and the exhaust gases from the high temperature annealing (residual Mo source, S source) are extracted through negative pressure pump. As can be seen from Fig. R24, the quartz tubes after high temperature annealing are as clean as brand-new quartz tubes and will not interfere with the next growth.

To address this comment, we have added the above discussion and Fig. R24 into Supporting Information as Supplementary Note 17 (page 17, lines 269-278).

Figure R24. The photographs of brand-new quartz tubes, the quartz tube after growth and the quartz tube after annealing (previously used one).

6. In line 288, the authors assert that they achieved record-high mobility of 140 cm²V⁻¹s⁻¹ on a transistor with dimensions of 1μm/5μm. This value appears consistent with the short-channel device. The authors list mobility of 140 cm²V⁻¹S⁻¹ only for the short channel devices. Please confirm these figures and elucidate why the mobility of the long-channel matches that of the short-channel devices.

Response 22:

We appreciate the Reviewer's comment. For the mobility of 140 cm²V⁻¹s⁻¹, it is extracted from the *h*-BN encapsulated device with channel length/width of 5 μm / 1 μm. For our 200 nm short-channel device, the extracted mobility is ~76.5 cm²V⁻¹s⁻¹. The lower mobility in short-channel device than *h*-BN encapsulated long-channel device

can be understood that in short-channel devices the contact resistance has a much larger weight in the total resistance.

7. Please indicate the thickness of the hBN layers (both top and bottom) for the hBN-encapsulated devices.

Response 23:

Sorry for the lack of information on the thickness of the *h*-BN layers. Figure R25a (R25c) shows the atomic force microscopy (AFM) image of the top (bottom) *h*-BN layer. The corresponding height profiles (Figs. R25b and R25d) indicate that the thickness of top and bottom *h*-BN layers is ~27 nm and ~34 nm, respectively.

To address this comment, we have added a sentence “The thickness of top and bottom *h*-BN layers is ~27 nm and ~34 nm, respectively (Supplementary Fig. 19)” in the revised manuscript (page 9, lines 291-292) and added Fig. R25 into Supporting Information as Supplementary Fig. 19 (page 14).

Figure R25. (a,b) The AFM image of the top *h*-BN layer (a) and its corresponding height profiles (b). (c,d) The AFM image of the bottom *h*-BN layer (c) and its corresponding height profiles (d).

8. As the transistor devices utilize dual gates for both long-channel hBN-encapsulated devices and short-channel HfO₂ devices, please measure the gate capacitances for both devices.

Response 24:

We appreciate the Reviewer’s comment. Figures R26a and R26b show a metal/*h*-BN/metal structure for capacitance measurement. The effective area between top and bottom metal electrodes is $20 \times 33 + 5 \times 3 \mu\text{m}^2$; the thickness of *h*-BN is ~54 nm (Fig. R26c). The measured capacitance is $\sim 382 \times 10^{-15} \text{F}$. Then we can extract the

dielectric constant of *h*-BN $\epsilon_r = \frac{C \cdot d}{\epsilon_0 \cdot S} = \frac{382 \times 10^{-15} \text{F} \times 54 \text{nm}}{8.854 \times 10^{-12} \text{F/m} \times (20 \times 33 + 5 \times 3) \mu\text{m}^2} = 3.452$, which is in good agree with previous results [*Adv. Electron. Mater.* 6, 2000550 (2020); *npj 2D Mater. Appl.* 2, 6 (2018)]. Based on the measured dielectric constant of *h*-BN, we can obtain the capacitance per unit area for top gate ($C_i = \frac{\epsilon_0 \cdot \epsilon_r}{d} = \frac{8.85 \times 10^{-12} \text{F/m} \times 3.45}{27 \times 10^{-9} \text{m}}$) = $1.131 \times 10^{-7} \text{F/cm}^2$) and bottom gate ($C_i = \frac{8.85 \times 10^{-12} \text{F/m} \times 3.45}{34 \times 10^{-9} \text{m}}$) = $0.872 \times 10^{-7} \text{F/cm}^2$).

For our short-channel HfO₂ devices, we employ the single-gate structure. Figure R27 is the C-V curves with the measurement frequency of 1 kHz, and amplitude of 50 mV.

The capacitance per unit area at V=4 V is $C_i = \frac{1.15 \times 10^{-12} \text{F}}{100 \times 10^{-6} \times 100 \times 10^{-6} \text{m}^2} = 1.15 \times 10^{-6} \text{F/cm}^2$.

To address this comment, we have added Fig. R26 and Fig. R27 into Supporting Information as Supplementary Fig. 20 (page 14) and Supplementary Fig. 23 (page 16).

Figure R26. (a) Optical image of a metal/*h*-BN/metal structure. The effective area between top and bottom metal electrodes is $20 \times 33 + 5 \times 3 \mu\text{m}^2$. (b,c) The thickness of *h*-BN is ~ 54 nm measured by AFM. (d) C-V curves. The measured capacity is 382 fF.

Figure R27. The gate capacitances for the short-channel HfO_2 devices.

We thank the Reviewer for his/her very valuable reviewing efforts. We hope we have convincingly addressed all the comments raised by the Reviewer that our revised manuscript meets the criteria for publication in *Nature Communications*.

REVIEWER COMMENTS

Reviewer #1 (Remarks to the Author):

The authors have satisfactorily addressed my concerns in the revised version and rebuttal letter. A reasonable mechanism about the unidirectional epitaxy that a buffer layer may be the key role has been proposed. Although the buffer layer has been observed and discussed by many groups and still deserves more in-depth investigations in the future, the novelty and creditability of the current work are not doubted. I suggest its publication in Nature Communications.

Reviewer #2 (Remarks to the Author):

The authors have diligently and effectively addressed the majority of the concerns and comments previously raised. In light of this thorough revision and the significant contributions their work offers to the field, I recommend its publication in Nature Communication.

Reviewer #3 (Remarks to the Author):

The authors have addressed numerous questions with comprehensive details. However, there appears to be a fundamental shift in the manuscript's focus from kinetic control to buffer layer control. It is important that the authors should provide sufficient evidence to support their points of view.

1. Could the authors provide the lattice structures of the buffer layer? Given the significance of this aspect, I strongly recommend that the authors present potential lattice structures supported by experimental results from Scanning Tunneling Microscopy (STM) or theoretical predictions from Density Functional Theory (DFT) or Molecular Dynamics (MD) simulations, although their initial assertion of such analysis being outside the current work's scope.
2. The manuscript should detail the evolution of the buffer layer corresponding to variations in the Mo/S ratio, along with its influence on the unidirectional orientation percentage of the MoS₂ layers. For instance, at a Mo/S ratio of 2.1%, one would expect to observe a fully continuous buffer layer in STEM cross-sectional imagery, whereas at a Mo/S ratio of 2.9%, a discontinuous or absent buffer layer might be anticipated. Please provide such cross-sectional images as evidence.
3. It would be very helpful for readers to provide Scanning Tunneling Microscopic (STM) images showing the Moiré patterns formed between MoS₂ and the reconstructed sapphire layer, which are critical for corroborating the discussions on interfacial interactions within the manuscript.

Point-by-point response to the Reviewers' comments

Reviewer #1 (Remarks to the Author):

The authors have satisfactorily addressed my concerns in the revised version and rebuttal letter. A reasonable mechanism about the unidirectional epitaxy that a buffer layer may be the key role has been proposed. Although the buffer layer has been observed and discussed by many groups and still deserves more in-depth investigations in the future, the novelty and creditability of the current work are not doubted. I suggest its publication in *Nature Communications*.

Response 1:

We sincerely thank the Reviewer for the positive evaluation on our revisions and his/her recommendation for publication in *Nature Communications*.

Reviewer #2 (Remarks to the Author):

The authors have diligently and effectively addressed the majority of the concerns and comments previously raised. In light of this thorough revision and the significant contributions their work offers to the field, I recommend its publication in *Nature Communication*.

Response 2:

We sincerely thank the Reviewer for the positive evaluation on our revisions and his/her recommendation for publication in *Nature Communications*.

Reviewer #3 (Remarks to the Author):

The authors have addressed numerous questions with comprehensive details. However, there appears to be a fundamental shift in the manuscript's focus from kinetic control to buffer layer control. It is important that the authors should provide sufficient evidence to support their points of view.

Response 3:

We sincerely thank the Reviewer for the positive evaluation on our responses and the insightful comments to improve our manuscript.

1. Could the authors provide the lattice structures of the buffer layer? Given the significance of this aspect, I strongly recommend that the authors present potential lattice structures supported by experimental results from Scanning Tunneling Microscopy (STM) or theoretical predictions from Density Functional Theory (DFT) or Molecular Dynamics (MD) simulations, although their initial assertion of such analysis being outside the current work's scope.

Response 4:

We sincerely thank the Reviewer's insightful comments. Since STM measurements require the conductive substrate while our sapphire substrate is insulating, it is impossible to perform the STM measurements. Notably, X-ray photoelectron spectroscopy (XPS) can offer a feasible technology to uncover the key information about the lattice structures of the buffer layer [e.g., *ACS Nano* 17, 10010 (2023)]. Figure R1 shows the high-resolution XPS spectrum of Mo 3d, after removing the as-grown unidirectional MoS₂ by a non-destructive water-assisted technique. A deconvolution and curve fitting reveal that the Mo 3d spectrum can be decomposed into a doublet corresponding to 3d_{5/2} Mo⁵⁺ and 3d_{3/2} Mo⁵⁺.

Figure R1. The high-resolution XPS spectrum of Mo 3d.

Based on the XPS results and in conjunction with previous results of MoS₂ epitaxy on β -Ga₂O₃ (001) [e.g., *ACS Nano* 17, 10010 (2023)], we infer that one possible configuration of the buffer layer is O–Mo–O–Al, with Mo exhibiting a (+5) oxidation state. To determine the detailed structure of the O–Mo–O–Al buffer layer, we further perform DFT calculations (as kindly suggested by the Reviewer) with the Vienna ab initio software package. The Perdew-Burke-Ernzerhof (PBE) exchange-correlation functionals are employed, along with DFT-D2 method of Grimme van der Waals corrections. An energy cutoff of 520 eV for the plane wave basis sets and a Γ -centered k-mesh of $1 \times 1 \times 1$ is used for geometry optimization and electronic structure calculations. The convergence of total energy with respect to these parameters is examined and found to reach the level of less than 0.001 eV/f.u. To prevent artificial interactions between periodic slab images, a vacuum thickness greater than 15 Å was applied. The Al₂O₃ (001) surfaces are modeled using large supercell slabs containing 9 atomic layers. The bottom 5 layers of atoms are kept fixed at their optimized bulk positions and other atomic positions are fully optimized with the lattice constant for the

supercell kept fixed at a value corresponding to the experimental lattice constant for a 1×1 unit cell.

Figure R2 shows the calculated the structure of the O–Mo–O–Al buffer layer on Al_2O_3 surface after fully structural relaxation. Generally, every O atom under the Mo layer is connected to two Al and two Mo atoms, and every Mo atom is connected to 3 O atoms under it. Meanwhile, every O atom above the Mo layer is connected to two Mo atoms and 3 O atoms are assigned to 1 Mo atom. Further, MoS_2 triangles with different orientations are positioned on the surface of the Al_2O_3 substrate with O–Mo–O–Al buffer layer. The DFT results reveal that the 60° structure, compared to the 0° structure, has a lower free energy of 110 meV, demonstrating a preferred growth orientation for MoS_2 triangles.

Figure R2. The atomic structures of MoS_2 growth on $\alpha\text{-Al}_2\text{O}_3$ (001) with buffer layer. The side view of the relaxed MoS_2 layer in 0° (a) and 60° (b) directions grown on $\text{Mo}^{+5}/\text{Al}_2\text{O}_3$ substrate and the corresponding top view of MoS_2 on the buffer layer. The Mo, S, O and Al atoms are shown in purple, yellow, red, and blue respectively.

To address this comment, we have added a new section (Supplementary Note 9. The structure of the buffer layer) into the Supporting Information which includes the above discussion, and Figs. R1 and R2 have been added into Supplementary Note 9 as Supplementary Figs. 13 and 14 (pages 9 and 10). In addition, we added a sentence “By performing X-ray photoelectron spectroscopy and density functional theory calculation, we infer that one possible configuration of the buffer layer is O–Mo–O–Al, with Mo exhibiting a (+5) oxidation state (Supplementary Note 9)” into the revised manuscript (page 5).

2. The manuscript should detail the evolution of the buffer layer corresponding to variations in the Mo/S ratio, along with its influence on the unidirectional orientation percentage of the MoS₂ layers. For instance, at a Mo/S ratio of 2.1%, one would expect to observe a fully continuous buffer layer in STEM cross-sectional imagery, whereas at a Mo/S ratio of 2.9%, a discontinuous or absent buffer layer might be anticipated. Please provide such cross-sectional images as evidence.

Response 5:

We appreciate the Reviewer's kind suggestions. Figure R3 show the cross-sectional HAADF-STEM images of a MoS₂ grown on the α -Al₂O₃ (0001) substrate along the α -Al₂O₃ $\langle 10\bar{1}0 \rangle$ direction, where the degree of unidirectional alignment is ~ 0 . No buffer layer below the as-grown MoS₂ is observed. This is in stark contrast to the case of unidirectional MoS₂ grown on the α -Al₂O₃ (0001) substrate where an atomically thin buffer layer is formed below the as-grown MoS₂.

To address this comment, we have added a sentence "By contrast, no buffer layer is formed below the as-grown MoS₂ is observed when the degree of unidirectional alignment is ~ 0 (Supplementary Fig. 11)" in the revised manuscript (page 4). Meanwhile, Figure R3 has been added to Supporting Information as Figure S11 (page 8).

Figure R3. Cross-sectional HAADF-STEM images of a MoS₂ grown on the α -Al₂O₃ (0001) substrate along the α -Al₂O₃ $\langle 10\bar{1}0 \rangle$ direction, where the degree of unidirectional alignment is ~ 0 .

3. It would be very helpful for readers to provide Scanning Tunneling Microscopic (STM) images showing the Moiré patterns formed between MoS₂ and the reconstructed sapphire layer, which are critical for corroborating the discussions on interfacial interactions within the manuscript.

Response 6:

Because STM measurements require the conductive substrate while our sapphire substrate is insulating, we cannot perform the STM measurements. We sincerely thank the Reviewer for the kind suggestion.

We thank the Reviewer for his/her very valuable reviewing efforts. We hope we have convincingly addressed all the comments raised by the Reviewer that our revised manuscript meets the criteria for publication in *Nature Communications*.

REVIEWERS' COMMENTS

Reviewer #3 (Remarks to the Author):

The authors have effectively addressed the majority of my previous concerns, significantly enhancing the manuscript's quality. Their work offers notable contributions to the field of single-crystal MoS₂ synthesis. Therefore, I recommend accepting this manuscript for publication in Nature Communications.